# Improved inference of population histories by integrating genomic and epigenomic data

**Thibaut Sellinger[1,2], Frank Johannes[3], Aurélien Tellier[1]\***

[1]Professorship for Population Genetics, Department of Life Science Systems, Technical University of Munich, Munich, Germany; [2]Department of Environment and Biodiversity, Paris Lodron University of Salzburg, Salzburg, Austria; [3]Professorship for Plant Epigenomics, Department of Molecular Life Sciences, Technical University of Munich, Freising, Germany

**\*For correspondence:**
aurelien.tellier@tum.de

**Abstract** With the availability of high-quality full genome polymorphism (SNPs) data, it becomes feasible to study the past demographic and selective history of populations in exquisite detail. However, such inferences still suffer from a lack of statistical resolution for recent, for example bottlenecks, events, and/or for populations with small nucleotide diversity. Additional heritable (epi) genetic markers, such as indels, transposable elements, microsatellites, or cytosine methylation, may provide further, yet untapped, information on the recent past population history. We extend the Sequential Markovian Coalescent (SMC) framework to jointly use SNPs and other hyper-mutable markers. We are able to (1) improve the accuracy of demographic inference in recent times, (2) uncover past demographic events hidden to SNP-based inference methods, and (3) infer the hyper-mutable marker mutation rates under a finite site model. As a proof of principle, we focus on demographic inference in *Arabidopsis thaliana* using DNA methylation diversity data from 10 European natural accessions. We demonstrate that segregating single methylated polymorphisms (SMPs) satisfy the modeling assumptions of the SMC framework, while differentially methylated regions (DMRs) are not suitable as their length exceeds that of the genomic distance between two recombination events. Combining SNPs and SMPs while accounting for site- and region-level epimutation processes, we provide new estimates of the glacial age bottleneck and post-glacial population expansion of the European *A. thaliana* population. Our SMC framework readily accounts for a wide range of heritable genomic markers, thus paving the way for next-generation inference of evolutionary history by combining information from several genetic and epigenetic markers.

## eLife assessment

This **important** study extends existing sequentially Markovian coalescent approaches to include the combined use of SNPs and hypervariable loci such as epimutations. This is an intriguing addition to infer population size history in the recent past, and the authors provide **solid** validation of their methods via simulation and analysis of empirical data in *Arabidopsis thaliana*. Given the increasing availability of such data, this work is a timely contribution and represents a foundation for further developments to explore when and where these methods will be best used.

## Introduction

A central goal in population genetics is to reconstruct the evolutionary history of populations from patterns of genetic variation observed in the present. Relevant aspects of these histories include past

**Figure 1.** Schematic distribution of two markers along the genealogy and four genomes. (**A**) Schematic distribution of marker 1 (yellow star) and marker 2 (green star) along the genealogies in a sample of four genomes, both following a homogeneous Poisson process. (**B**) The green marker 2 is not heritable so its distribution is independent of the genealogy. (**C**) The green marker 2 is spatially structured along the genome, violating the distribution of the Poisson process along the genome and conflicting with the genealogy. (**D**) The green marker 2 does not follow the Poisson process through time, for example burst of mutations at a specific time point represented by given branches of the genealogies in green. The yellow marker 1 has an identical Poisson process along the genome and the genealogy in all four panels, and for readability, marker 2 exhibits light and dark green states.

The online version of this article includes the following figure supplement(s) for figure 1:

**Figure supplement 1.** Probability of a site to be segregating in a sample of size two for different mutation rates.

demographic changes as well as signatures of selection. Inference methods based on deep learning (DL, *Korfmann et al., 2023*), approximate Bayesian computation (ABC, *Boitard et al., 2016*), or sequential Markovian coalescent (SMC, *Li and Durbin, 2011*; *Schiffels and Durbin, 2014*) aim to infer this information directly from full genome sequencing data, which is becoming rapidly available for many (non-model) species due to decreasing costs. The SMC, in particular, offers an elegant theoretical framework that builds on the classical Wright-Fisher and the backward-in-time Kingman coalescent stochastic models (e.g. *Kingman, 1982*; *Charlesworth and Charlesworth, 2010*; *Wakeley, 2008*). Both models conceptualize Mendelian inheritance as generating the genealogy of a population (or a sample), that is, the unique history of a fragment of DNA passing from parents to offspring. When this genealogy includes the effect of recombination, it is called the ancestral recombination graph (ARG, *Hudson, 1983*; *Wiuf and Hein, 1999*).

Under the Kingmann coalescent model, the true genealogy of a population (or sample) is defined by its topology and branch length, and contains the information on past demographic changes and life history traits (*Nordborg, 2000*; *Sellinger et al., 2020*; *Strütt et al., 2023*; *Tellier et al., 2011*) as well as selective events (*Charlesworth and Charlesworth, 2010*; *Wakeley, 2008*). The genealogical and the mutational processes of any heritable marker can, therefore, be disentangled, and the frequency of any given marker state is given by the shape of the genealogy in time (see *Figure 1A*). A central assumption about heritable genomic markers is that they are generated by two homogeneous Poisson mutation processes along the genome as well as through time. This entails that mutations in different genealogies are independent due to the effect of recombination (*Wiuf and Hein, 1999*; *McVean and Cardin, 2005*), and that there are no time periods with a large excess, or a severe lack, of mutations along a genealogy (mutations are independently distributed in time within a DNA fragment). In other words, the frequencies of polymorphisms at DNA markers observed across a sample of sequences are constrained by, as well as inform on, the underlying genealogy at this locus (*Figure 1A*). To clarify these assumptions, we present a schematic representation of marker 1 (yellow in *Figure 1*), which fulfils both homogeneous Poisson processes in time and along the genome. We

also present cases applicable to a second genomic marker 2 that violates the model assumptions, namely by not being heritable (*Figure 1B*) or not following a non-homogeneous Poisson process in the genome (*Figure 1C*) or in time (*Figure 1D*).

Despite the power of the SMC, well-known model violations such as variation in recombination and mutation rates along the genome (*Barroso et al., 2019*; *Barroso and Dutheil, 2023*) or pervasive selection (*Schraiber and Akey, 2015*; *Johri et al., 2021*; *Johri et al., 2020*) can compromise the accuracy of demographic and selective inference (*Gattepaille et al., 2013*; *Sellinger et al., 2021*). There are two other important issues that have received less attention in the literature. The first issue occurs when the population recombination rate ($\rho$) is higher than the population mutation rate (θ). In such cases, inferences can be biased if not erroneous (*Terhorst et al., 2017*; *Sellinger et al., 2021*; *Sellinger et al., 2020*), because several recombination events cannot be inferred due to the lack of single-nucleotide polymorphisms (SNPs for point mutations). This problem affects many species, though interestingly not humans which have a ratio $\rho/\theta \approx 1$. A second issue occurs when the mutational process along the genealogy is too slow to be informative about sudden and strong variation in population size (i.e. population bottlenecks), such as during colonization events of novel habitats. The typical low mutation rate of $10^{-9}$ up to $10^{-8}$ (per base, per generation) found in most species, therefore, places strong limitations on SMC analysis of recent bottleneck events (up to ca. $10^4$ generations ago) when inference is based solely on SNP data. Indeed, bottlenecks are often either not found, or when inferred, their timing and magnitude are not well estimated (inferred smoother than in reality, *Johri et al., 2021*; *Sellinger et al., 2021*), even when a large number of samples is used. A typical example is the large uncertainty of the timing and magnitude of the population size bottleneck during the last glacial maximum (LGM) and post-LGM expansion in *Arabidopsis thaliana* European populations based on several studies using different accessions and SMC inference methods (*Alonso-Blanco et al., 2016*; *Durvasula et al., 2017*).

Nonetheless, current SMC, DL, or ABC inference methods making use of full genome sequence data rely almost exclusively on SNPs for inference (*Schiffels and Durbin, 2014*; *Terhorst et al., 2017*; *Sellinger et al., 2020*; *Boitard et al., 2016*; *Korfmann et al., 2024*). There are both practical and theoretical reasons for using SNPs: They are easily detectable from short-read re-sequencing data and their mutational process is well approximated by the infinite site model (*Charlesworth and Charlesworth, 2010*; *Wakeley, 2008*), simplifying the inference of the underlying genealogy. However, other heritable genomic markers exist whose mutation rates can be several orders of magnitude higher than that of SNPs and could thus be more informative about recent demographic events. These include microsatellites, insertions, deletions, and transposable elements (TEs). Although those heritable markers are not necessarily neutral (such as TEs, which are likely to be under weak purifying selection), they contain information on the evolutionary history of the population. Current technological limitations still impede the easy detection and estimation of allele frequencies for many of these markers (*Yang et al., 2018*; *Ou et al., 2019*; *Wang, 2018*). For example, identifying insertion/excision variation of transposable elements or copy number variation of microsatellites requires a high-quality reference genome and ideally long-read sequencing approaches (*Ou et al., 2019*). In addition to these genomic markers, DNA cytosine methylation is emerging as a potentially useful epigenetic marker for phylogenetic inference in plants (*Yao et al., 2021*; *Yao et al., 2023*). Stochastic gains and losses of DNA methylation at CG dinucleotides, in particular, arise at a rate of ca. $10^{-4}$ up to $10^{-3}$ per site per generation (that is 4–5 orders of magnitude faster than DNA point mutations, *van der Graaf et al., 2015*), and can be inherited across generations (*Pisupati et al., 2023*; *Weigel and Colot, 2012*). These so-called spontaneous epimutations are likely neutral at the genome-wide scale (*Vidalis et al., 2016*; *Johannes and Schmitz, 2019b*, but see *Muyle et al., 2021*; *Pisupati et al., 2023*), and can be easily detected from bisulphite converted short read sequencing data (*Lister et al., 2008*; *Schmitz et al., 2013*). Recent work suggests that CG methylation data can be used as a molecular clock for timing divergence between pairs of lineages over timescales ranging from years to decades (*Yao et al., 2023*).

However, the theoretical integration of the above-mentioned (epi)genomic markers into a population genomics and SMC inference framework is not trivial. Because of the high mutation rate, the mutational process at these (hyper-mutable) markers is reversible and more consistent with a finite site, rather than infinite site, model, which can result in extensive homoplasy (as known for microsatellite markers, *Estoup et al., 2002*). Indeed, classic expectations of population genetics diversity

statistics, mostly built for SNPs, need to be revised for these hyper-mutable markers (*Charlesworth and Jain, 2014*; *Wang and Fan, 2014*). Here, we develop the theoretical and methodological inference framework named SMCtheo to include additional (potentially hyper-mutable) markers in the SMC. We showcase our model using extensive simulations as well as application to published DNA cytosine methylation data (in genic regions) from local populations of *A. thaliana* (*Schmitz et al., 2013*; *Vidalis et al., 2016*). We demonstrate that integration of hyper-mutable genomic markers into SMC models significantly improves the inference accuracy of past variation of population size or can even uncover demographic events not uncovered using SNPs alone. Our proof-of-principle approach opens up novel avenues for studying population genetic processes over time scales that have been largely inaccessible using traditional SNP-based approaches. This may prove particularly useful when exploring recent demographic changes of endangered species as a way to assess their potential for extinction in the context of biodiversity loss and global change.

## Results

### Theoretical results with two markers underlying the SMC computations

We study polymorphic sites across genomes of several sampled individuals which exhibit several possible markers (DNA nucleotides, methylation, TEs, indels, microsatellites,…). We define any marker by (1) its maximum number of possible states ($nb_s$), for example nucleotide sites have four states (A, T, C and G) while a methylation site has two states (methylated or unmethylated), and (2) its mutation rate μ, that is the rate at which the state of a marker changes into another state per position and per generation *Anzai et al., 2003*; for simplicity, we assume equal mutation rates between all bases, known as the Jukes-Cantor model. More specifically, we are interested in two rates: the DNA mutation rate for changes in DNA nucleotides and the epimutation rate for changes in methylation state. Furthermore, we assume that at each position on the genome, only one type of marker can occur and be observed. We obtain as a first theoretical result the probability for a given site in the genome to be identical ($P(id)$) or segregating ($P(seg)$) (i.e. polymorphic) in a sample of size two ($n = 2$, two sampled chromosomes are compared):

$$P(id, n = 2) = \frac{1}{nb_s} + \frac{(nb_s - 1)}{nb_s} e^{-2\mu t_M \frac{(nb_s)}{(nb_s - 1)}}$$

$$P(seg, n = 2) = \frac{(nb_s - 1)}{nb_s} - \frac{(nb_s - 1)}{nb_s} e^{-2\mu t_M \frac{(nb_s)}{(nb_s - 1)}}$$

(1)

This probability is a function of the time to the most recent common ancestor (TMRCA in text and $t_M$ in *Equation 1*, details in Appendix 1 and 2 B). The probability for a mutation to occur for a given marker increases with an increased TMRCA (*Charlesworth and Charlesworth, 2010*; *Wakeley, 2008*), but under high mutation rates (and high effective population size) the marker may not be polymorphic in the sample as mutations may be reversed (so-called homoplasy, *Estoup et al., 2002*; *Charlesworth and Jain, 2014*). In *Figure 1—figure supplement 1*, we illustrate these properties by computing the probability in *Equation 1* for different mutation rates. The inference of recent demographic events and bottlenecks relies on the presence of polymorphic sites to detect recent coalescent events (TMRCA), and should be improved by using markers with high (or fast) mutation rate (e.g. hyper mutable).

In the following, we simulate data under different demographic scenarios using the sequence simulator program *msprime* (*Baumdicker et al., 2022*; *Kelleher et al., 2016*), which generates the ARG of $n$ sampled diploid individuals (set to $n = 5$ throughout this study, leading to 10 haploid genomes). This ARG contains the genealogy of a given sample at each position of the simulated chromosomes. We then process the ARG to create DNA sequences according to the model parameters and the type of marker considered. We first assume a set of genomic markers obtained for a sample size $n$ and mutating according to a homogeneous Poisson process along the genome and in time (along the genealogy) as in *Figure 1A*. To simulate the sequence data, we define the number of marker types (any number between 1 and the sequence length) and the proportion of sites of each marker type in the sequence. Each marker is characterized by both parameters $nb_s$ and μ. For simplicity, we simulate sequences with two markers but note that the method can be easily extended to additional markers.

**Table 1.** Average estimated mutation rate of the second theoretical genomic marker.
Average estimated values of the mutation rate of marker 2 ($\mu_2$), knowing that of marker 1. We use 10 sequences (5 diploid individuals) of 100 Mb ($r = \mu_1 = 10^{-8}$ per generation per bp) under a constant population size fixed at $N = 10,000$. The coefficient of variation over 10 repetitions is indicated in parentheses.

| True $\mu_2$ value | Estimated value of $\mu_2$ |
|---|---|
| $10^{-8}$ | $9.9 \times 10^{-9}$ (0.02) |
| $10^{-6}$ | $1.0 \times 10^{-6}$ (0.008) |
| $10^{-4}$ | $1.4 \times 10^{-4}$ (0.01) |
| $10^{-2}$ | $3.05 \times 10^{-3}$ (0.41) |

Marker 1 represents 98% of the sequence and has a per-site mutation rate $\mu_1 = 10^{-8}$ mimicking nucleotide SNP markers under an infinite site model (thus considered as bi-allelic at a given DNA site, *Yang, 1996*). By contrast, marker 2 composes the complementary 2% of the sequence length, with a per-site mutation rate of $\mu_2 = 10^{-4}$ per generation between two possible states. Marker 2 is thus hyper-mutable compared to marker 1 and mimics methylation/epimutation sites. Note that mutation events at Markers 1 and 2 are simulated under a finite site model.

We use different SMC-based methods throughout this study. These methods include: (1) MSMC2 used as a reference method (*Malaspinas et al., 2016*), (2) SMCtheo is an extension of the PSMC' (*Li and Durbin, 2011*; *Schiffels and Durbin, 2014*) accounting for any number of heritable theoretical markers, and (3) eSMC2 which is equivalent to SMCtheo but accounting only for SNPs markers (*Sellinger et al., 2021*) (to avoid any bias in implementation differences between SMCtheo and MSMC2). All methods are hidden Markov models (HMM) derived from the pairwise sequentially Markovian coalescent (PSMC') (*Schiffels and Durbin, 2014*) and assume neutral evolution and a panmictic population. The hidden states of these methods are the coalescence time of a sample of size two at a position on the sequence. From the distribution of the hidden states along the genome, all methods can infer population size variation through time as well as the recombination rate (*Schiffels and Durbin, 2014*; *Malaspinas et al., 2016*; *Sellinger et al., 2021*).

## The inclusions of hyper-mutable genomic markers improves demographic inference

We assume that the mutation rate of marker 1 is $\mu_1 = 10^{-8}$ per generation per bp. We use this information to estimate the mutation rate of marker 2, which we vary from $\mu_2 = 10^{-8}$ to $\mu_2 = 10^{-2}$ per generation per bp. The estimation results based on simulated data under a constant population size of $N = 10,000$ are displayed in *Table 1*. We find that our approach is capable of inferring $\mu_2$ with high accuracy for rates up to $\mu_2 = 10^{-4}$. However, when the mutation rate $\mu_2$ is $10^{-2}$, our approach underestimates it by a factor three, suggesting the existence of an accuracy limit. To demonstrate that information can be gained by integrating marker 2 (with $\mu_2 = 10^{-4}$), we compared the ability of several inference methods to recover a recent bottleneck (*Figure 2A*). All methods correctly infer the amplitude of population size variation. When accounting only for marker 1 (with $\mu_1 = 10^{-8}$), MSMC2 and eSMC2 fail to infer accurately the sudden variation of population size. However, with the inclusion of hyper-mutable marker 2, our SMCtheo approach correctly infers the rapid change of population size of the bottleneck (*Figure 2A*, green). It is encouraging that an accurate estimation of the demography is obtained, even when the mutation rate of marker 2 is unknown (*Figure 2A*, blue).

Furthermore, some species or populations might feature small effective population sizes (ca. $N = 1,000$), potentially resulting in reduced genomic diversity. In such cases, the inclusion of hyper-mutable markers should also improve demographic inference. We present the results of such a scenario in *Figure 2A*, where the population size was divided by a factor 10 compared to the previous scenario in *Figure 2A*. We find that in the absence of the hyper-mutable marker 2, no approach can correctly infer the variation of population size. From the shape of the inferred demography, methods using only marker 1 do not suggest the existence of a bottleneck followed by recovery (the 'U-shaped' demographic scenario is not apparent with the orange and red lines, *Figure 2B*). Yet, when integrating

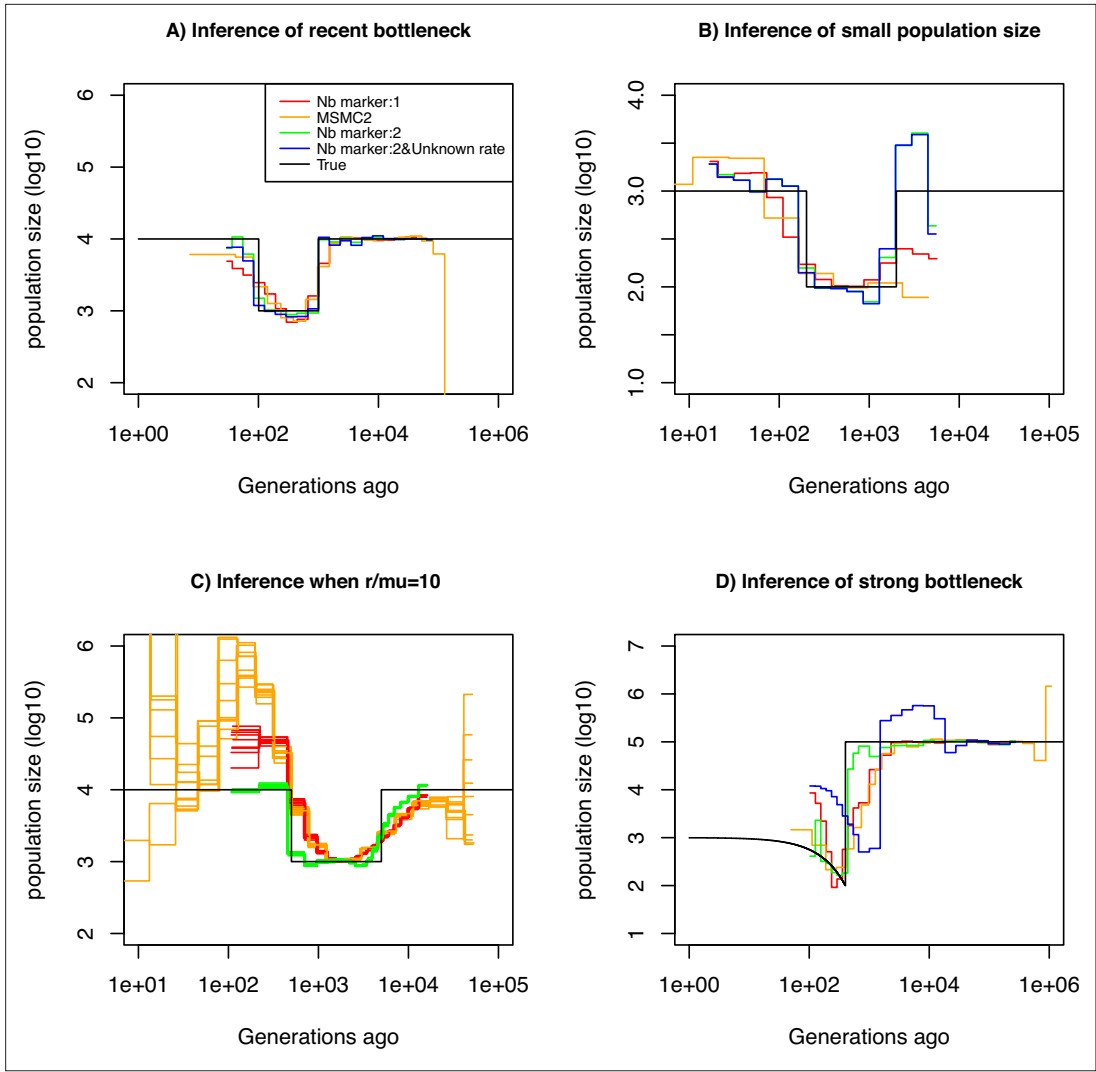

**Figure 2.** Performance of SMC approaches using different markers. Estimated demographic history of a bottleneck (black line) by SMC approaches using two genomic markers. In orange and red are the estimates by MSMC2 and eSMC2 based on only marker 1. Estimates from SMCtheo integrating both markers are in green (with known $\mu_2$) and in blue with unknown $\mu_2$. The demographic scenarios are (**A**) 10-fold recent bottleneck with an ancestral population size $N = 10,000$, (**B**) 10-fold recent bottleneck with an ancestral population size $N = 1,000$, (**C**) 10-fold bottleneck with an ancestral population size $N = 10,000$, and (**D**) a very severe (1000 fold) and very recent bottleneck with incomplete size recovery. In A, B, and D, we assume $r/\mu_1 = 1$ (with $r = \mu_1 = 10^{-8}$, $\mu_2 = 10^{-4}$ per generation per bp) and in C, $r/\mu_1 = 10$ (with $r = 10^{-7}$, $\mu_1 = 10^{-8}$, and $\mu_2 = 10^{-4}$ per generation per bp). In all cases (**A, B, C and D**) 10 sequences (5 diploid indivudals) of 100 Mb were used as input.

The online version of this article includes the following figure supplement(s) for figure 2:

**Figure supplement 1.** Performance of of SMCtheo using two theoretical markers when marker 2 is very rare.

**Figure supplement 2.** Performance of the SMCtheo using theoretical markers by maximizing the true Likelihood function.

---

both markers, the population size can be recovered, even if the mutation rate of marker 2 is not a priori known. In both *Figure 2A and B*, we assume that marker 2 occurs at a frequency of 2% in the genome. This percentage may be unrealistically high depending on the marker and the species. To test the impact of reducing marker 2 frequency, we repeat the simulations shown in *Figure 2A* but set its frequency to as low as 0.1% (a 20-fold reduction). We find that the inclusion of the hyper-mutable marker 2 continues to improve inference accuracy in very recent times, albeit less pronounced than in *Figure 2A* (see *Figure 2—figure supplement 1*). This suggests that a very small proportion of hyper-mutable genomic sites is sufficient to significantly improve the accuracy of inferences.

All full genome inference methods, especially SMC approaches, display lower accuracy when the population recombination rate ($\rho = 4Nr$) is larger than the population mutation rate of marker

**Table 2.** Estimates of recombination rates with one or both markers.
For SMCtheo, BW stands for the use of the Baum-Welch algorithm to infer parameters, and LH for the use of the likelihood. We use 10 sequences of 100 Mb with $r = 10^{-7}$, $\mu_1 = 10^{-8}$ and $\mu_2 = 10^{-4}$ per generation per bp in a population with a past bottleneck event. The coefficient of variation over 10 repetitions is indicated in brackets.

| Method | True recombination rate | Average estimated recombination rate |
|---|---|---|
| MSMC2 (BW) | $10^{-7}$ | $0.23 \times 10^{-7}$ (0.017) |
| 1 Marker: BW | $10^{-7}$ | $0.25 \times 10^{-7}$ (0.012) |
| 2 Marker: BW | $10^{-7}$ | $0.90 \times 10^{-7}$ (0.004) |
| 1 Marker: LH | $10^{-7}$ | $0.84 \times 10^{-7}$ (0.036) |
| 2 Marker: LH | $10^{-7}$ | $0.94 \times 10^{-7}$ (0.01) |

1 ($\theta_1 = 4N\mu_1$). We simulate sequence data under a bottleneck scenario slightly more ancient than in **Figure 2A** and assume that $\rho/\theta_1 = r/\mu_1 = 10$ and $\rho/\theta_2 = r/\mu_2 = 10^{-3}$. Our results show that by integrating the genomic marker 2 which mutation rate is larger than the recombination rate, estimates of the recombination rate as well as past population size variation are substantially improved (**Table 2**, **Figure 2C**). Indeed, analyzing only marker 1, eSMC2 and MSMC2 identify the bottleneck (albeit smoothed) and only slightly overestimate recent population size (**Figure 2D**). By integrating the hyper-mutable marker 2, our SMCtheo approach correctly infers the strength and time of the bottleneck when $\mu_1$ and $\mu_2$ are known (**Figure 2D**, green line), while the timing of the bottleneck is slightly shifted in the past when $\mu_2$ is unknown and estimated by our method (**Figure 2D**, blue line). When $\mu_2$ is unknown, SMCtheo additionally infers a spurious sudden variation of population size between 10,000 and 100,000 generations ago. Using only marker 1, the estimates of the recombination rate are inaccurate (**Table 2**). To complete the visual representation and provide a quantitative assessment of inference accuracy, we compute the root mean square error (RMSE) values for demographic inference (**Supplementary file 1a**). We further improve the accuracy of estimation by optimizing the likelihood (LH) to estimate the recombination rate and demography compared to the classically used Baum-Welch (BW) algorithm (**Table 2** and **Figure 2—figure supplement 1**, **Figure 2—figure supplement 2**). Our results demonstrate that SNPs are limiting and insufficient for accurate inferences in recent times and that the inclusion of an additional marker with a mutation rate higher than the recombination rate generates significant improvements in demographic inference. However, by directly optimizing the likelihood, the true recombination rate can be recovered well even with marker 1 only.

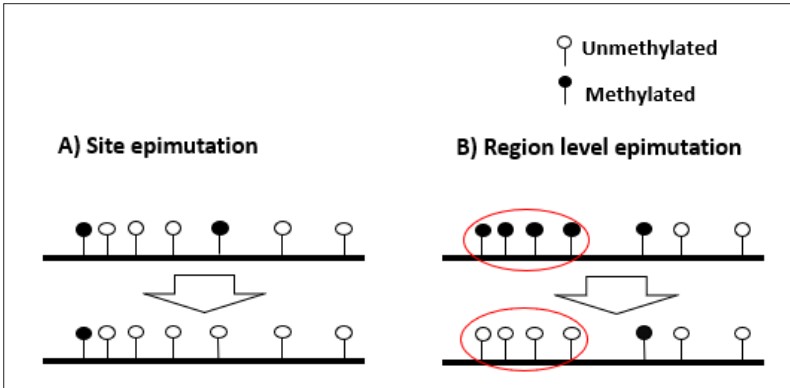

**Figure 3.** Schematic representation of site and region epimutations. Schematic representation of a sequence undergoing epimutation at (**A**) the cytosine site level and (**B**) at the region level. A methylated cytosine in CG context is indicated in black, and an unmethylated cytosine in white.

## Integrating DNA methylation improves the accuracy of inference

### Definition of the theoretical model for DNA methylation

Following the previously encouraging results of demographic inference with SNPs and a hyper-mutable marker under the specific assumptions of *Figure 1A*, we develop a specific SMCm method to jointly analyse SNPs and CG methylation as an epigenetic hyper-mutable marker. Since our SMCm stems from the eSMC (*Sellinger et al., 2020*; *Strütt et al., 2023*), it corrects for the effect of self-fertilization when applying to *A. thaliana*. We focus here on methylation located in CG contexts within genic regions as these have been found to evolve neutrally (*Vidalis et al., 2016*; *Yao et al., 2021*; *Yao et al., 2023*). The methylation of individual CG dinucleotides produces a biallelic heritable marker with a finite number of (epi)mutable sites (*Figure 3*). In a sample of several sequences from a population, variation in the methylation status of individual CGs is known as single methylation polymorphism (SMP, *Figure 3A*), which could be used for demographic and divergence inference (*van der Graaf et al., 2015*; *Vidalis et al., 2016*). However, CG methylation sites can also be organized in spatial clusters (of similar state) due to region-level epimutation (*Figure 3B*, *Weigel and Colot, 2012*; *Denkena et al., 2021*; *Muyle et al., 2021*). Region-level epimutations can have different epimutation rates than individual CG sites. Population-level variation in the methylation status of these clusters is known as differentially methylated regions (DMRs). Furthermore, when integrating SMP and DMR epimutational processes (i.e. what we here call region-level epimutation), the methylation status of CG sites is therefore affected by the superposition of both processes. Therefore, the simulation and modeling of epimutation processes of SMPs are more complex than in our previous model as we need to account for the effect of region methylation as well as for methylation and demethylation epimutation rates to be different and asymmetrical (*van der Graaf et al., 2015*; *Denkena et al., 2021*).

To make our simulations realistic, we use the *A. thaliana* genome sequence as a starting point, and focus on CG dinucleotides within genic regions. To that end, we select random 1 kb regions within genes and choose only those CG sites that are clearly methylated or unmethylated in *A. thaliana* natural populations based on whole genome bisulphite sequencing (WGBS) measurements from the 1001 G project (SI text). Our simulator for CG methylation is built in a similar way as the one described above, but the epimutation rates are allowed to be asymmetric with the per-site methylation rate ($\mu_{SM}$) and demethylation ($\mu_{SU}$). Region-level epimutations are also implemented, setting the region length to either 1 kb (*Muyle et al., 2021*) or 150 bp (*Denkena et al., 2021*). The region-level methylation and demethylation rates are defined as $\mu_{RM}$ and $\mu_{RU}$, respectively. We assume that site-level and region-level epimutation processes are independent. Making this assumption explicit later allows us to test if it is violated in comparisons with actual data. Our simulator also assumes that DNA mutations and epimutations are independent of one another. That is, for simplicity, we ignore the fact that methylated cytosines are more likely to transition to thymines as a result of spontaneous deamination (*Johannes, 2019a*). We also ignore the possibility that new DNA mutations could act as CG methylation quantitative trait loci and affect CG methylation patterns in both cis and trans. Such events are extremely rare, so the above assumptions should hold reasonably well over short evolutionary time scales. As the goal is to apply our approach to *A. thaliana*, we simulate sequence data for a sample size $n = 10$ (but considering *A. thaliana* haploid) from a population displaying 90% selfing (*Sellinger et al., 2020*) under a recent severe population bottleneck demographic scenario. We simulate data assuming previous estimates of the rates of recombination (*Salomé et al., 2012*), DNA mutation (*Ossowski et al., 2010*), and site- and region-level methylation (*van der Graaf et al., 2015*; *Denkena et al., 2021*).

As guidance for future analyses of demographic inference using SNPs and DNA methylation data, the theoretical and empirical analysis of *A. thaliana* methylomes consist of the following five steps: (1) assessing the relevance of region-level methylation (DMRs) for inference, (2) inference of site and region epimutation rates, (3) comparing statistics for the SNPs, SMPs, and DMRs distributions, (4) demographic inference using SNPs with SMPs or DMRS, and (5) demographic inference using SNPs with SMPs and DMRs.

### Step 1: assessing the relevance of region-level methylation (DMRs) for inference

We determine our ability to detect the existence of spatial correlations between epimutations. That is, we asked if site-specific epimutations can lead to region-level methylation status changes across a

range of epimutation rates (assuming two sequences of 100 Mb, $r = \mu_1 = 10^{-8}$ per generation per bp and a constant population size $N = 10,000$, results in *Supplementary file 1b*). If site-specific epimutations are independently distributed, the probability of a given site being in a given (methylated or unmethylated) state should be independent of the state of nearby sites (knowing the epimutation rate per site). Conversely, if there is a region effect on epimutation (DMRs), two consecutive sites along the genome would exhibit a positive correlation in their methylated states. We therefore calculate from the per-site (de)methylation rates $\mu_{SM}$ and $\mu_{SD}$ the probability that two successive cytosine positions are identical in their methylation assuming they are independent. This probability can be compared to the one observed from methylation data (here simulated) so that we obtain a statistical test for the existence of a positive correlation in the methylation status of nearby sites, interpreted as a regional-level epimutation process (p-value = 0.05) according to *Figure 1A*. A small p-value of the test (<0.05) suggests the existence of a region effect for methylation/demethylation affecting neighbouring cytosines, contrary to a high p-value indicating no spatial structure of methylation distribution. We find that when region epimutation rates are higher than (or similar to) site-level epimutation rates, namely $\mu_{RM} \gtrapprox \mu_{SM}$ and ($\mu_{RU} \gtrapprox \mu_{SU}$), the existence of regions of consecutive cytosines is detected with high accuracy. However, when site-level epimutation rates are higher ($\mu_{SU} > \mu_{RU}$ and $\mu_{SM} > \mu_{RM}$) than region-level epimutation rates, region-level changes cannot be readily detected (*Supplementary file 1b*). When methylated regions are detected, we can further determine their length using a specifically developed HMM using all pairs of genomes (similarly to *Shahryary et al., 2020*; *Denkena et al., 2021*; *Taudt et al., 2018*). While the length of the methylated region is pre-determined in our simulations (1 kb or 150 bp), site-level epimutation occur which can change the distribution of methylation states in that region and across individuals, thus DMR regions can vary in length along the genome and between pairs of chromosomes.

## Step 2: inference of site- and region-level epimutation rates

As the epimutation rates of most plant species remain unknown, we assess the accuracy of SMCm to infer epimutation rates at the site- and region-level directly from simulated data. We first assume that either only site- or only region epimutations can occur and infer their respective rates (see *Supplementary file 1c and d*). Our SMCm approach can accurately recover these rates except when these are higher than 10⁻⁴. Next, we assess the accuracy of our approach to simultaneously infer site- and region-level epimutation rates, assuming that region and site epimutation rates are equal (*Supplementary file 1e* and *Figure 4—figure supplement 1*). Similar to our previous observation, we find that when the epimutation rates are very high (e.g. close to 10⁻²), accuracy is lost compared to slower epimutation rates. Nonetheless, our average estimated rates are off from the true value by less than a factor 10. Hence, under our model assumptions, we can recover the correct order of magnitude for site- and region-level methylation and demethylation rates.

## Step 3: distribution of statistics for SNPs, SMPs, and DMRs

To gain insights on the distribution of epimutations under the described assumptions, we look at key statistics from our simulations: the distribution of distance between two recombination events versus the distribution of the length of estimated DMR regions (*Figure 4A*), and the LD decay for SMPs (in genic regions) and SNPs (in all contexts) (*Figure 4C and D*). In our simulations, DMR regions have a maximum fixed size, but their length depends on the interaction between the region- and site-level epimutation rates. As mentioned in step 1, the methylated/demethylated regions are detected using the binomial test and their length estimated by the HMM. Therefore, while variation exists for the length of these regions (*Figure 4A*), regions are on average shorter than the span of genealogies along the genome, which are defined by the frequency of recombination events along the genome ($r = 3.5 \times 10^{-8}$ as in *A. thaliana*). There is virtually no linkage disequilibrium (LD) between epimutations due to the high epimutation rate (*Figure 4C*), while the LD between SNPs can range over few kbp (*Figure 4D*, as observed in *A. thaliana* [*Cao et al., 2011*; *Schmitz et al., 2013*]). Note, however, that the region methylation process in itself does not generate LD because this measure can only be computed if SMPs are present in frequency higher than $2/n$ in the sample, that is there is no LD measure defined for monomorphic methylated/unmethylated regions. In other words, our simulator generates SNPs, SMPs, and DMRs, which fulfil the three key assumptions from *Figure 1A*. We note that by using a constant population size $N = 10,000$, the LD decay for SNPs is higher than in the *A.*

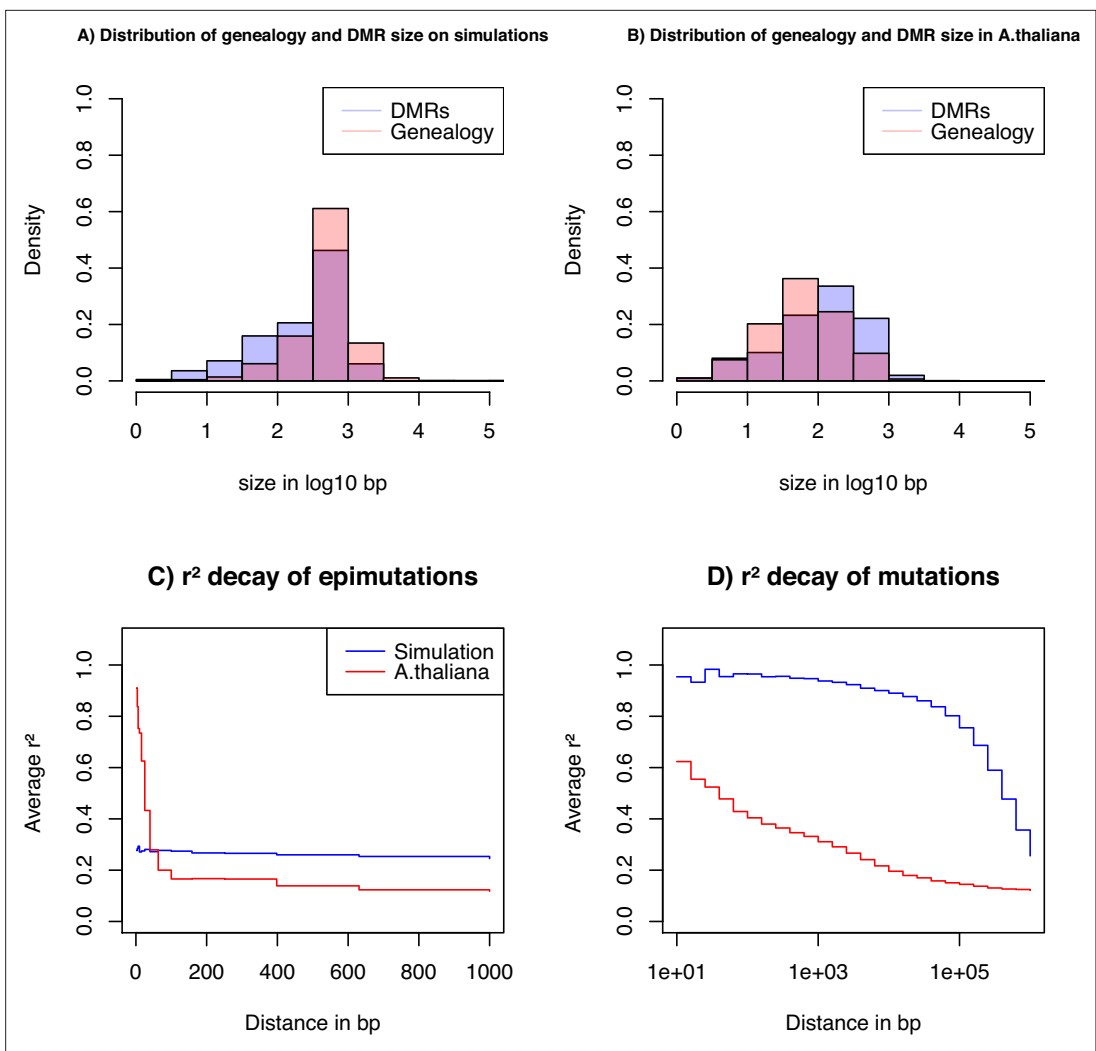

**Figure 4.** Key statistics for epimutations and mutations. (**A**) Histogram of the length between two recombination events (genomic span of a genealogy) and DMRs size in bp of the simulated data. (**B**) Histogram of genealogy span and DMRs size in bp from the *A. thaliana* data (10 German accessions). (**C**) Linkage disequilibrium decay of epimutations in our samples of *A. thaliana* (red) and simulated data (blue). (**D**) Linkage disequilibrium decay of mutations in our *A. thaliana* samples (red) and simulated data (blue). The simulations reproduce the outcome of a recent bottleneck with sample size $n = 5$ diploid of 100 Mb, the rates per generation per bp are $r = 3.5 \times 10^{-8}$, $\mu_1 = 7 \times 10^{-9}$, $\mu_{SM} = 3.5 \times 10^{-4}$, $\mu_{SU} = 1.5 \times 10^{-3}$, and per 1 kb region $\mu_{RM} = 2 \times 10^{-4}$ and $\mu_{RU} = 1 \times 10^{-3}$.

The online version of this article includes the following figure supplement(s) for figure 4:

**Figure supplement 1.** Average estimates of the site and region methylation and demethylation rates for simulated data.

---

*thaliana* data which exhibit an effective population size of ca. $N = 250,000$ (*Cao et al., 2011*) and past changes in size.

## Step 4: demographic inference based on SNPs with SMPs or DMRs

We test the usefulness of either SMPs or DMRs for demographic inference. Simulations under the demographic model from steps 1–3 assume DNA mutations (SNPs) and only site epimutations (SMPs), that is no region-level methylation ($\mu_{RM} = \mu_{RU} = 0$). We perform inference of past demographic history under different amounts of potentially methylated sites with and without a priori knowledge of the methylation/demethylation rates (*Figure 5A and B*). When the site epimutation rates are a priori known, the sharp decrease of population size can be accurately detected. When epimutation rates are unknown, the shape of the past demographic history is also well inferred except for a scaling issue (a shift along the x- and y-axes similar to that in *Figure 5D*). When we vary the amount of potentially

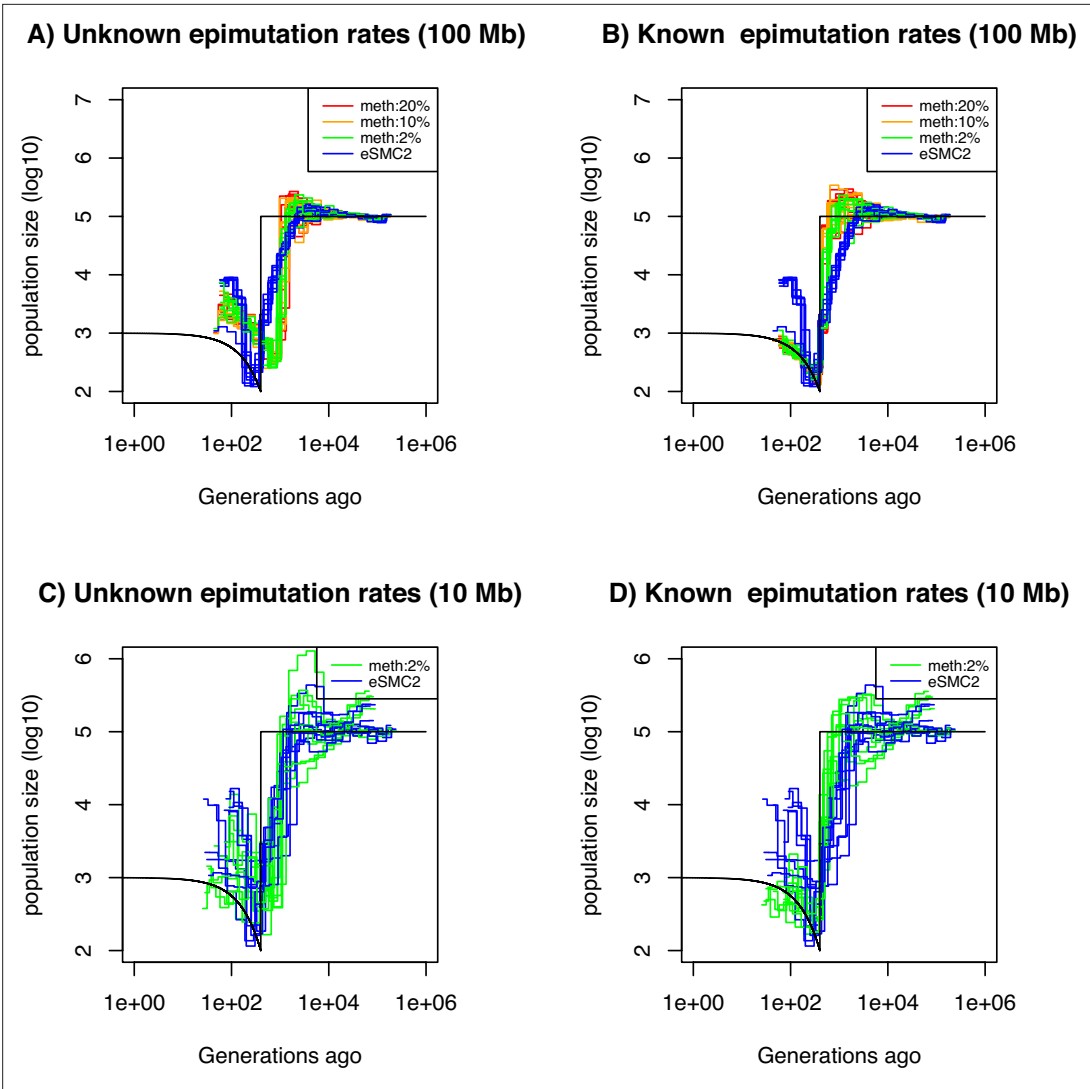

**Figure 5.** Performance of SMC approaches using site epimutations (SMPs) and mutations (SNPs) under a bottleneck scenario. Estimated demographic history by eSMC2 (blue) and SMCm assuming the epimutation rate is known (**B and D**) or not (**A and C**) where the percentage of CG sites with methylated information varies between 20% (red), 10% (orange) and 2% (green) using 10 sequences of 100 Mb in **A and B** (with 10 repetitions) and 10 sequences of 10 Mb in **C and D** (three repetitions displayed) under a recent severe bottleneck (black). The parameters are: $r = 3.5 \times 10^{-8}$ per generation per bp, mutation rate $\mu_1 = 7 \times 10^{-9}$, methylation rate to $\mu_{SM} = 3.5 \times 10^{-4}$ and demethylation rate to $\mu_{SU} = 1.5 \times 10^{-3}$ per generation per bp.

The online version of this article includes the following figure supplement(s) for figure 5:

**Figure supplement 1.** Performance of SMCm for methylation with only DMR regions of length 1kbp.

**Figure supplement 2.** Performance of SMCm for methylation with only DMR regions of length 150 bp.

**Figure supplement 3.** Performance of SMCm for methylation with site and region epimutations.

**Figure supplement 4.** Performance of SMCm for methylation, accounting only for SMPs.

**Figure supplement 5.** Average pvalue of the binomial test for epimutations.

methylated sites (2%, 10%, and 20%) our inference results remain largely unchanged. This suggests that having methylation measurements for as low as 2% of all CG sites being epimutable in the genome is entirely sufficient to improved SNP-based demographic inference (eSMC2 in **Figure 5A**). The RMSE values for demographic inference are computed for all cases in **Figure 5** to provide an additional quantitative understanding of our results (**Supplementary file 1f**).

The amount of sequence data used in **Figure 5A, B** is fairly large compared to real datasets (10 haploid genomes of length 100 Mb). We therefore ran the SMCm and eSMC2 on sequence data

simulated under the same scenario but with a reduced sequence length of 10 Mb ($n = 5$ diploid, *Figure 5C and D*, only 3 repetitions are presented for visibility). In this case, we found that inference is significantly affected when using only SNPs (eSMC2 in blue), as we are unable to correctly recover the demographic scenario. However, incorporating SMPs with known site-level epimutations into the model leads to substantial inference improvements (*Figure 5C, D*, *Supplementary file 1f*). We additionally quantify the accuracy gain in ARG inference by inferring the expected coalescent time (TMRCA) at each position in the genome by the three approaches (eSMC2, SMCm with unknown epimutation rates and SMCm with known epimutation rates) under the same scenario from *Figure 5*. The RMSE values of the TMRCA inference are presented in *Supplementary file 1g*. We confirm our intuition that integrating epimutations slightly improves the accuracy of TMRCA when the epimutation rates are known but does not when the rates are unknown.

To quantify the effect of DMRs on inference, we simulate data under the same demographic scenario but assume only region-level epimutations (DMRs, $\mu_{SM} = \mu_{SU} = 0$). The results for DMR region sizes 1 kb and 150 bp are displayed in *Figure 5—figure supplement 1* and *Figure 5—figure supplement 1*, respectively. As in *Figure 5*, we observed a gain of accuracy in inference when region-level epimutation rates are known, while the length of the region (1 kb or 150 bp) does not seem to affect the result. However, no significant gain of information is observed when integrating DMR data with unknown epimutation rates (*Figure 5—figure supplement 1*, *Figure 5—figure supplement 2*). In summary, CG methylation SMPs and to a lesser extend DMRs, can be used jointly with SNPs to improve demographic inference (*Supplementary file 1h* presents the corresponding RMSE values for demographic inference shown in *Figure 5—figure supplement 1*, *Figure 5—figure supplement 2*), especially in recent times (*Supplementary file 1f and h*).

## Step 5: demographic inference based on SNPs with SMPs and DMRs

Since site- and region-level methylation processes can occur in real data, we run SMCm on simulated data under the same demographic scenario, but now using both site (SMPs) and region (DMRs) epimutations and accounting for both mutation processes (with rates similar to the one found in *A. thaliana*). Inference results are displayed in *Figure 5—figure supplement 3* (RMSE values in *Supplementary file 1i*). When the epimutations rates are unknown, we observe a gain of accuracy when integrating epimutations, especially in recent times. However, when epimutation rates are a priori known, we observe a loss of accuracy when accounting for epimutations. This loss of accuracy is due to the mislabeling of the methylation region status (in step 1) when site and region-level epimutations occur jointly at similar rates (as there will be methylated sites in unmethylated regions and unmethylated sites in methylated regions).

Finally, we assess the inference accuracy when using SNPs and SMPs but ignoring in SMCm the region methylation effect (DMRs), even though this latter process takes place (*Figure 5—figure supplement 4*, RMSE values in *Supplementary file 1j*). The inference accuracy decreases compared to the previous results (*Figure 5—figure supplements 1–4*), and while the sudden variation of population is somehow recovered, the estimates of the time and magnitude of size change are not well recovered in recent times. Hence, those results demonstrate the importance of accounting for site and region-level epimutations processes in steps 1–5. We demonstrate that our SMCm can exhibit, to some extent, an improved statistical power for demographic inference using SNPs and SMPs while accounting for site and region-level methylation processes under the assumptions of *Figure 1A*. We show that (1) using SMPs we can unveil past demographic events hidden by limitations in SNPs, (2) the correct demography can be uncovered irrespective of knowing a priori the epimutation rates, (3) ignoring site or region-level processes can decrease the accuracy of inference, and (4) knowing the epimutation rates may improve the estimate of demography compared to simultaneously estimating them with SMCm.

## Joint use of SNPs and SMPs improves the inference of recent demographic history in *A. thaliana*

### Step 1: assessing the strength of region-level methylation process in *A. thaliana*

We apply our inference model to genome and methylome data from 10 *A. thaliana* plants from a German local population (*Cao et al., 2011*). We start by assessing the strength of a region effect on

the distribution of methylated CG sites along the genome. As expected from *Denkena et al., 2021*, for all 10 individual full methylomes, we reject the hypothesis of a binomial distribution of methylated and unmethylated sites along the genomes, suggesting the existence of region effect methylation (yielding DMRs), meaning that CG are more likely to be methylated if in a highly methylated region, and conversely for unmethylated CG. This is consistent with the autocorrelations in mCG found in *Denkena et al., 2021*; *Briffa et al., 2023*; *Lyons et al., 2023*. As a first measure of methylated region length, we test the independence between two annotated CG methylations given a minimum genomic distance between them (within one genome). We observe an average p-value smaller than 0.05 for distances up to 2,000 bp but then the p-value rapidly increases (>0.4) (*Figure 5—figure supplement 5*). As a second measure, our HMM (based on pairs of genomes) yields a DMR average length of 222 bp (distribution in *Figure 4B*).

We conclude that the minimum distance for epimutations to be independent along a genome is over 2 kb and spans larger distances than the typically proposed DMR size (ca. 150 bp in *Denkena et al., 2021* and 222 bp in our analysis) and can therefore cover the size of a gene (see *Muyle et al., 2021*; *Briffa et al., 2023*). The simulations and data from *A. thaliana* indicates that the epimutation processes that produce DMRs at the population level in plants cannot simply result from the cumulative action of single-site epimutations. This insight is consistent with recent analyses of epimutational processes in gene bodies, which seems to indicate that the autocorrelation in CG methylation is a function of cooperative methylation maintenance and the distribution of histone modifications (*Briffa et al., 2023*; *Lyons et al., 2023*).

## Step 2: site- and region-level epimutation rates

We use the rates empirically estimated in *A thaliana* and taken in the above simulations ($\mu_{SM} = 3.5 \times 10^{-4}$ and $\mu_{SU} = 1.5 \times 10^{-3}$ per bp per generation and $\mu_{RM} = 2 \times 10^{-4}$ and $\mu_{RU} = 1 \times 10^{-3}$ per region per generation, *van der Graaf et al., 2015*; *Denkena et al., 2021*).

## Step 3: distribution statistics for SNPs, SMPs and DMRs in *A. thaliana*

Since our SMC model assumes that DNA, SMP and DMR polymorphisms are determined by the underlying population/sample genealogy, DMR which span long genomic regions may spread across multiple genealogies and thus violates our modeling assumptions. We thus further investigate the potential discrepancies between the data and our model (*Figure 4*). We infer the DMR sizes from all 10 *A. thaliana* accessions using our ad hoc HMM, and measure the bp distance between a change in the expected hidden state (i.e. coalescent time) along the genome, which we interpret as recombination events (called the genomic span of a genealogy). The resulting distributions are found in *Figure 4B*. We observe that both distributions have a similar shape, but DMRs are, on average, twice as large as the inferred genomic genealogy span: average length of 222 bp (DMR) vs 137 bp (genealogy) and median length of 134 bp (DMR) vs 62 bp (genealogy). This means that, on average, DMRs are larger than the average distance between two recombination events, thus violating the homogeneous distribution of epimutations along the genome (*Figure 1C*).

To further unveil potential non-homogeneity of the distribution of epimutations, we assess the decay of LD of mutations (SNPs) and epimutations (SMPs) (*Figure 4C and D*), confirming the results in *Schmitz et al., 2013*. We find the LD between SMPs in the data to be high (and higher than LD between SNPs) for distances smaller than 100 bp (red line in *Figure 4C, D*). The LD decay of SMPs is much faster than for SNPs (no linkage disequilibrium between epimutations for distances >100 bp), likely stemming from (1) epimutation rates being much higher than the DNA mutation rate, and (2) the high per-site recombination rate in *A. thaliana*. Moreover, the LD between SMPs at distances smaller than 100 bp in *A. thaliana* being much higher compared to our simulations (*Figure 4C*), we suggest that additional local mechanisms of epimutation processes may not be accounted for in our model of the region-level methylation process.

## Step 4: demographic inference for *A. thaliana* based only on SNPs and SMPs

Finally, we apply the SMCm approach to data from the German accessions of *A. thaliana*. When using SNP data only, the demographic results are similar to those previously found (*Sellinger et al., 2020*; *Strütt et al., 2023*; *Figure 6* purple lines), with no strong evidence for an expansion post-Last Glacial Maximum (LGM) (*Cao et al., 2011*). We then sub-sample and analyze segregating SMPs, which exhibit

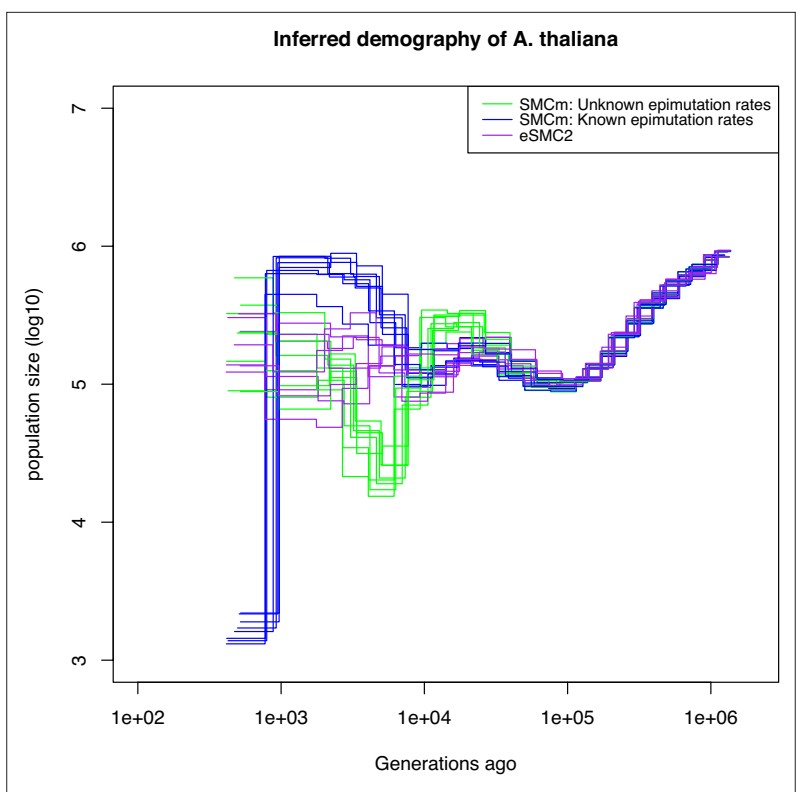

**Figure 6.** Integrating epimutations and mutations on German accessions of *A. thaliana*. Estimated demographic history of the German population by eSMC2 (only SNPs, purple) and SMCm when keeping polymorphic methylation sites (SMPs) only: green with epimutation rates estimated by SMCm, blue with epimutation rates fixed to empirical values. The region epimutation effect is ignored. The parameters are $r = 3.6 \times 10^{-8}$, $\mu_1 = 6.95 \times 10^{-9}$, and when assumed known, the site methylation rate is $\mu_{SM} = 3.5 \times 10^{-4}$ and demethylation rate is $\mu_{SU} = 1.5 \times 10^{-3}$.

The online version of this article includes the following figure supplement(s) for figure 6:

**Figure supplement 1.** Demographic estimation using all methylation sites from German accessions of *A. thaliana*.

**Figure supplement 2.** Average number of segregating site per window of 100kp on chromosome 1.

**Figure supplement 3.** Average number of segregating site per window of 100kp on chromosome 2.

**Figure supplement 4.** Average number of segregating site per window of 100kp on chromosome 3.

**Figure supplement 5.** Average number of segregating site per window of 100kp on chromosome 4.

**Figure supplement 6.** Average number of segregating site per window of 100kp on chromosome 5.

both methylated and unmethylated states in our sample (as in *van der Graaf et al., 2015*). Here we ignore DMRs and account only for SMPs. When we use as input the methylation and demethylation rates that have been inferred experimentally (*van der Graaf et al., 2015*), a mild bottleneck post-LGM is followed by recent expansion (*Figure 6* blue lines). By contrast, letting our SMCm estimate the epimutation rates, we find in recent times a somehow similar but stronger demographic change post-LGM. We find a strong bottleneck event occurring between ca. 5,000 and 10,000 generations ago, followed by an expansion until today (*Figure 6* green lines). The inferred site epimutation rates are 10,000 faster than the DNA mutation rate (*Supplementary file 1k*), which is close to the expected order of magnitude from experimental measures with and without DMR effects (*van der Graaf et al., 2015*; *Denkena et al., 2021*). Both estimates thus yield a post-LGM bottleneck followed by a recent population expansion. These results indicate that the inclusion of DNA methylation data can aid in the accurate reconstruction of the evolutionary history of populations, particularly in the recent past, where SNPs reach their resolution limit. This is made possible by the fact that the DNA methylation status at CG dinucleotide undergoes stochastic changes at rates that are several orders of magnitude higher than the DNA mutation rate and can be inherited across generations similar to DNA mutations.

## Step 5: demographic inference correcting for DMRs in *A. thaliana*

To assess the robustness of our inference results, we run SMCm using all cytosines (CG) sites with an annotated methylation status (segregating or not) while accounting or not for DMRs (*Figure 6—figure supplement 1*). We fix epimutation rates to the empirically estimated values and confirm the estimates from *Figure 6*. When the region-level methylation process is ignored, the inferred demography (blue lines in *Figure 6—figure supplement 1*) is similar to the estimates from SMPs with fixed rates in *Figure 6* (blue lines). When the region-level methylation process is taken into account (orange lines in *Figure 6—figure supplement 1*), the inferred demography is similar to that of *Figure 6* (green lines). In the case where we infer the epimutation rates (sites and region) the demographic history inference is not improved compared to that estimated using SNPs only (*Figure 6—figure supplement 1*, green and red lines) while the inferred epimutation rates are lower than expected (*Supplementary file 1k and l*), but the ratio of site to region epimutation rates is consistent with empirical estimates (*Denkena et al., 2021*).

## Discussion

Current approaches analyzing whole genome sequences rely on statistics derived from the distribution of ancestral recombination graphs (*Gattepaille et al., 2016*; *Sellinger et al., 2021*; *Korfmann et al., 2024*; *Strütt et al., 2023*; *Brandt,, 2022*; *Wohns et al., 2022*; *Speidel et al., 2019*; *Kelleher et al., 2019*). In this study we present a new SMC method that combines SNP data with other types of genomic (TEs, microsatallites) and epigenomic (DNA methylation) markers. We focus mainly on the inclusion of genomic markers whose mutation rates exceed the DNA point mutation rate, as such (hyper-mutable) markers can provide increased temporal resolution in the recent evolutionary past of populations, and aid in the identification of demographic changes (e.g. population bottlenecks). We demonstrate that by integrating multiple heritable genomic markers, the population size variation in very recent time can be more accurately recovered (outperforming any other methods given the amount of data used in this study [*Terhorst et al., 2017*; *Speidel et al., 2019*]). Our results indicate that correctly integrating multiple genomic marker can improve TRMCA inference, which is becoming a field of high interest (*Korfmann et al., 2024*; *Hubisz et al., 2020*; *Mahmoudi et al., 2022*). Our simulations demonstrate that if the SNP mutation rate is known, the mutation rate of other markers can be recovered (under the condition that the marker follow all hypotheses described in *Figure 1*). Moreover, our method accounts for the finite site problem that arises at reversible (hyper-mutable) markers and/or where effective population size is high (*Tellier et al., 2011*; *Upadhya and Steinrücken, 2022*). Overall, the simulator and SMC methods presented here therefore pave the way for a rigorous statistical framework to test if a common ARG can explain the observed diversity patterns under the model hypotheses laid out in *Figure 1*. We find that comparisons of LD for different markers along the genome is a useful way to assess violations of our model assumptions.

As proof of principle, we apply our approach on data originating from whole genome and methylome data of *A. thaliana* natural accessions (focusing on CG context in genic regions, as in *Vidalis et al., 2016*; *Yao et al., 2021*; *Yao et al., 2023*). Indeed, *A. thaliana* presents the largest genetic and epigenetic data-set of high quality. Additionally, the methylation states in CG context has been proven mainly heritable and is well documented (*Denkena et al., 2021*; *Hazarika et al., 2022*; *van der Graaf et al., 2015*). We first investigate the distribution of epimutations along the genomes. Our model-based approach provides strong evidence that DMRs cannot simply emerge from spontaneous site-level epimutations that arise according to a Poisson processes along genome. Instead, stochastic changes in region-level methylation states must be the outcome of spontaneous methylation and demethylation events that operate at both the site- and region-level (as corroborated by *Pisupati et al., 2023*; *Briffa et al., 2023*; *Lyons et al., 2023*). Our epimutation model cannot fully describe the observed diversity of epimutations along the genome (*Denkena et al., 2021*), meaning that the epimutation processes may indeed be more complex than expected (*Denkena et al., 2021*; *Hazarika et al., 2022*; *Briffa et al., 2023*; *Lyons et al., 2023*). We observe non-independence between annotated methylation sites spanning genomic regions larger than the span of the underlying genealogy (determined by recombination events) which no model can currently describe. Additionally, we find high LD between SMPs over short distances which does not appear in our simulations (simulation performed under the current measures of epimutation rates). Thus, methylation probably violate the

assumptions of a Poisson process distribution along the genome and in time (*i.e.* **Figure 1**), in line with recent functional studies (**Pisupati et al., 2023**; **Hazarika et al., 2022**; **Lyons et al., 2023**). We thus further caution against conclusions on the role of natural (purifying) selection (**Muyle et al., 2021**) or its absence (**Vidalis et al., 2016**) based on population epigenomic data due to the violation of the above-mentioned assumptions. Additionally, we suspect those model violations to explain the discrepancy between the epimutation rates we inferred and the ones measured experimentally (**van der Graaf et al., 2015**; **Denkena et al., 2021**). To solve this discrepancy, one would need to develop a theoretical epimutation model capable of describing the observed diversity at the evolutionary time scale and then use this model to reanalyse the sequence data from the biological experiment to re-estimate the epimutation rates. We thus suggest a possible way forward for modeling epimutations through an Ising model (**Zhang et al., 2018**) to account for the heterogeneous methylation process. However, our preliminary work and the simulation results in **Briffa et al., 2023**, indicate that such model generates non-homogeneous mutation process in space (i.e. along the genome) and time, violating our current SMC assumptions (**Figure 1C, D**). Hence, there is a need to develop a more realistic methylation model for epimutations. A model accounting for heterogeneous rates would probably need to rely on a more sophisticated HMM (*e.g.* continuous time Markov chains **Ki and Terhorst, 2023** for SMC approaches) than what is presented here or to use other full genome inference methods (see **Korfmann et al., 2024**) which are not constrained by the SMC assumptions (**Figure 1**) but depends on simulations.

Interestingly, the distance of LD decay for SMPs matches quite well the estimated distance between recombination events (**Figure 4**). In addition to our theoretical results in **Table 2**, this observation reinforces the usefulness of using SMPs (or any hyper-mutable marker) to improve estimates of the recombination rate along the genome in species where the per site DNA mutation rate (μ) is smaller than the per site recombination rate (*r*) as in *A. thaliana*.

Nonetheless, we find that a restricted focus on segregating SMPs in genic regions could meet our model assumptions reasonably well, and thus provides a promising way forward. Using these segregating SMPs, we recover a past demographic bottleneck followed by an expansion which could fit the post- Last Glacial Maximum (LGM) colonization of Europe (although caution must be taken concerning the reliability of those results as pointed above), a hypothesized scenario (**François et al., 2008**) which could not be clearly identified using SNPs only from European (relic and non-relic) accessions (**Cao et al., 2011**). Currently strong evidence from inference methods are lacking (**Cao et al., 2011**, Figure 4 in **Durvasula et al., 2017**). Indeed, beyond the limits of using SNPs only, current results are limited by theoretical frameworks unable to simultaneously account (and disentangle) for extensive background selection (reinforced by very high selfing), population structure and variation in molecular rates (e.g. mutation rates, **Monroe et al., 2022**), which are all known to be present in *A. thaliana*. Those various forces are known to bias inference results when non-accounted for (**Charlesworth and Jensen, 2023**; **Rodríguez et al., 2018**), and may explain the variance in our demographic estimates. We also note that using CG methylated sites in genic regions may be problematic as the typical genealogies at these loci could be shorter than the genome average due to the presence of background selection, thus making the inference of such short TMRCA more difficult (even with SMPs) than in non-coding regions (which do not harbour desirable CG methylation sites, **van der Graaf et al., 2015**; **Vidalis et al., 2016**; **Yao et al., 2021**).

We suggest that simultaneously accounting for multiple heritable markers can help disentangle different evolutionary forces, such as between selection and variation in mutation rate: selection has a local effect on the population genealogy, while the mutation rate variation would only locally affect that given marker but not the genealogy (**Charlesworth and Jensen, 2023**). The absence of conflicting demography inferred from SNPs and from methylation confirms at the time scale of thousands of generations, CG methylation sites are mainly heritable and can be modeled using population genetics theory (**Charlesworth and Jain, 2014**; **Vidalis et al., 2016** but see **Pisupati et al., 2023**) and used to estimate divergence between lineages (**Yao et al., 2023**; **Yao et al., 2021**). In other words, fast ecological local adaptation (**Schmid et al., 2018**) and response to stresses (**Srikant and Drost, 2020**) may likely not be prominent forces endlessly reshaping CG methylation patterns (non-heritability in **Figure 1B**).

Overall, our results demonstrate that our approach can be used in different cases. If the epimutations/genomic markers evolutionary mechanisms are not well understood (**Pisupati et al., 2023**;

*Briffa et al., 2023*; *Lyons et al., 2023*), our approach provides inference tools to study the markers' rates and distribution process along the genome, without requiring additional experimental data. If the evolution of epimutations/genomic markers are well understood (including a measure of the mutation rates) and can be modeled to describe the observed intra-population diversity, these can be integrated to improve the SMC performance. Hence, when applying our approach to genome-wide genetic and epigenetic data, it is advisable to use accurately annotated markers with, if possible, information regarding their inheritance and mutational properties. Regarding methylation specifically, while the set of gene body methylated genes previously used (*Vidalis et al., 2016*; *Yao et al., 2023*) are likely the optimal choice (*Yao et al., 2021*), these are too few and too scattered across the genome to maximize the statistical power of SMC methods. We, therefore, use methylation sites at all genic regions. Yet, despite the wealth of functional studies and data on methylation in *A. thaliana*, the distribution of epimutations is not fully understood (*Hazarika et al., 2022*; *Pisupati et al., 2023*), but independent rates for sites and region-level have been estimated (*van der Graaf et al., 2015*; *Denkena et al., 2021*; *Yao et al., 2023*). We note here the promising methylation modeling framework by *Briffa et al., 2023*; *Lyons et al., 2023*, albeit it does not yet consider evolutionary processes at the population level. Our results shed light on the inference accuracy in presence of site and region-level epimutations when occurring at similar rates (*Figure 5—figure supplement 3*). When accounting for region-level epimutations, our algorithm requires first inferring via an HMM the methylation status of a region in order to later on compute the epimutation probabilities (i.e. the emission matrix of the SMC HMM). Hence, in the presence of site and region-level epimutations occurring at similar rates, recovering the region methylation status becomes harder as methylated sites are observed in the unmethylated regions (and unmethylated sites observed in the methylated regions). The mislabelling of the region methylation status lead to accuracy loss due to the use of the wrong emission probability at the later steps of the SMC inference (Forward-Backward algorithm). When epimutation rates are freely inferred, their values are based on the estimated methylation region status. Therefore, even if the inferred rates are incorrect, these are sufficiently consistent with the inferred region methylation status to contain information and slightly improve inference accuracy. Additionally, extra care must be taken when dealing with epigenomic data in other species as the SMP calling might not be as simple as for *Arabidopsis thaliana* due to potential difference of methylation between different tissues or pool of cells. Similarly, we ignore here the potential dependence between SNPs and SMPs, as more empirical evidence (and modeling) is required to quantify the potential interaction between both mutational processes.

On a brighter note, with the release of new sequencing technology (*Lang et al., 2020*), long and accurate reads are becoming accessible, leading to the availability of highquality reference genomes for model and non-model species alike (*Nurk et al., 2020*). Additionally, the quality of re-sequencing (population sample) genome data and their annotations is enhanced so that additional markers such as transposable elements, insertion, deletion or microsatellites can be called with increasing confidence. These accurate genomes will provide access to new classes of genomic markers that span the entire mutational spectrum. We therefore suspect that, in the near future, there will be an improvement in our understanding of the heritability of many markers besides SNPs. Adding other genomic markers besides SNPs will improve full genome approaches, which are currently limited by the observed nucleotide diversity (*Kelleher et al., 2019*; *Speidel et al., 2019*; *Schweiger and Durbin, 2023*). Additionally, the potential complexity resulting by integrating multiple independent markers could be tackled by the use of continuous time Markov chains for the emission matrix. We predict that our results pave the way to improve the inference of (1) biological traits or recombination rate through time (*Deng et al., 2021*; *Strütt et al., 2023*), (2) multiple merger events (*Korfmann et al., 2024*), and (3) recombination and mutation rate maps (*Barroso et al., 2019*; *Barroso and Dutheil, 2023*). Our method also should help to dissect the effect of evolutionary forces on genomic diversity (*Johri et al., 2022*; *Johri et al., 2021*), and to improve the simultaneous detection, quantification and dating of selection events (*Albers and McVean, 2020*; *Bisschop et al., 2021*; *Johri et al., 2020*).

Hence, there is no doubt that extending our work, by simultaneously integrating diverse types of genomic markers into other theoretical framework (e.g. ABC approaches), likely represents the future of population genomics, especially to study species for which many thousands of samples cannot be obtained. We believe our approach helps to develop more general classes of models capable of

leveraging information from any type and amount of diversity observed in sequencing data, and thus to challenge our current understanding of genome evolution.

## Materials and methods

### Simulating two genomic markers

The sequence is written as a sequence of markers with a given state. Each site is annotated as M $X$ S $Y$, where $X$ indicates the marker type and $Y$ is the current state of that marker: for example, M1S1 indicates a marker of type 1 in state 1 at this position. To simulate sequence of theoretical marker we start by simulating an ARG which is then split in a series of genealogies (i.e. a sequence of coalescent trees) along the chromosome and create an ancestral sequence (based on equilibrium probability of marker states). Mutation events (nucleotides or epimutations for methylable cytosine) are added along the sequence, that is along the series of genealogies. The ancestral sequence is thus modified by mutation event assuming a finite site model (*Yang, 1996*) conditioned to the branch length and topology of the genealogies. Each leaf of the genealogy is one of the $n$ samples. Our model has thus two important features: (1) markers are independent from one another, and (2) a given marker has a polymorphism distribution between samples (frequencies of alleles) determined by one given genealogy. The simulator can be found in the latest version of eSMC2 R package (https://github.com/TPPSellinger/eSMC2; copy archived at *Sellinger, 2024a*).

### Simulating methylome data

We now focus on methylation data located at cytosine in CG context within genic regions. Only, CG sites in those regions are considered 'methylable', and CG sites outside those defined genic regions do not have a methylation status and are considered 'unmethylable'. We vary the percentage of CG site with methylation state annotated from 2 to 20% of the sequence length. The simulator can in principle simulate epimutations in different methylation context and different rates (*Lister et al., 2008*; *Cokus et al., 2008*; *Zilberman et al., 2007*; *Zhang et al., 2006*). We simulate epimutations as described above but with asymmetric rates: the methylation rate per site is $\mu_{SM} = 3.5 \times 10^{-4}$, and the demethylation rate per site is $\mu_{SM} = 1.5 \times 10^{-3}$(*van der Graaf et al., 2015*; *Denkena et al., 2021*). For simplicity and computational tractability, we assume that when an epimutation occurs, it occurs on both DNA strands which then present the same information. In other words, for a haploid individual, a cytosine site can only be methylated or unmethylated (as in *Taudt et al., 2018*). For region level epimutations, the region length is either 1kbp (*Muyle et al., 2021*) or 150 bp (*Denkena et al., 2021*). The region level methylation and demethylation rates are set to $\mu_{RM} = 2 \times 10^{-4}$ and $\mu_{RU} = 10^{-3}$ respectively (similar to rates measured in *A. thaliana*, *Denkena et al., 2021*). In addition to this, unlike for theoretical marker described above, mutations, site and region epimutations can occur at the same position of the sequence.

To simulate methylation data, we start with an ancestral sequence of random nucleotide and then randomly select regions in which CG sites have their methylation state annotated (representing the genic regions). Cytosine in CG context in those regions are either methylated or unmethylated (noted as M or U). Cytosine in other context or regions are considered as non-methylable (and noted as C). The ancestral methylation state is then randomly attributed according to the equilibrium probabilities. Our simulator then introduces DNA mutations, site- and region-epimutations in a similar way as described above.

### SMC methods

All three methods (eSMC2, SMCtheo, and SMCm) are based on the same mathematical foundations and implemented in a similar way within the eSMC2 R package (https://github.com/TPPSellinger/eSMC2; *Strütt et al., 2023*; *Korfmann et al., 2024*; *Sellinger et al., 2021*; copy archived at *Sellinger, 2024a*). This allows to specifically quantify the accuracy gained by accounting for multiple genomic markers.

### SMC optimization function

All current SMC approach rely on the Baum-Welch (BW) algorithm for parameter estimation in order to reduce computational load (as described in *Terhorst et al., 2017*). Yet, the Baum-Welch algorithm is

an Expectation-Maximization algorithm, and can hence fall in local extrema when optimizing the likelihood. We alternatively extend SMCtheo to estimate parameters by directly optimizing the likelihood (LH) at the greater cost of computation time (even when using the speeding techniques described in *Sand et al., 2013*). We run this approach on a sub-sample of size six haploid genomes to limit the required computational time.

## eSMC2 and MSMC2

SMC methods based on the PSMC' (*Schiffels and Durbin, 2014*), such as eSMC2 and MSMC2, focus on the coalescent events between two individuals (i.e. two haploid genomes or one diploid genome). The algorithm moves along the sequence and estimates the coalescence time at each position by assessing whether the two sequences are similar or different at each position. If the two sequences are different, this indicates a mutation took place in the genealogy of the sample. The intuition being that the absence of mutations (i.e. the two sequences are identical) is likely due to a recent common ancestor between the sequences, and the presence of several mutations likely reflects that the most recent common ancestor of the two sequences is distant in the past. In the event of recombination, there is a break in the current genealogy and the coalescence time consequently takes a new value according to the model parameters (*Marjoram and Wall, 2006*; *Schiffels and Durbin, 2014*). A detailed description of the algorithm can be found in *Malaspinas et al., 2016*; *Sellinger et al., 2020*.

## SMCtheo based on several genomic markers

Our SMCtheo approach is equivalent to PSMC' but takes as input a sequence of several genomic markers. The algorithm goes along a pair of haploid genomes and checks at each position which marker is observed and whether both states of the marker are identical or not. The approach is identical to the one described above, except that the probability of both sequences being identical at one site depends on the mutation rate of the marker at this site (*Equation 1*). While the mutation rates for many heritable genomic markers are unknown, there is an increasing amount of measures of the DNA (SNP) mutation rate for many species. Our SMCtheo approach is able to leverage the information from the distribution of one theoretical marker (e.g. mutations for SNPs) to infer the mutation rate of the other marker 2 (assuming both mutation rates to be symmetrical). If more than 1% of sites are polymorphic in a sequence, we use the finite site assumption. If not, then from the diversity observed, the different mutation rates can be recovered by simply comparing Waterson's theta ($\theta_W$) between the reference marker (i.e. with known rate) and the marker with the unknown rates. For example, if the diversity ($\theta_W$) at marker 2 is smaller by a factor ten than the reference marker 1 (and no marker violates the infinite site hypothesis), the mutation rate of marker 2 is inferred to be ten times smaller (corrected by the number of possible states). However, if the marker 2 violates the infinite site hypothesis, a Baum-Welch algorithm is run to infer the most likely mutation rates under the SMC to overcome this issue (the Baum-Welch algorithm description can be found in *Sellinger et al., 2020*).

## SMCm

When integrating epimutations, the number of possible observations increases compared to eSMC2. As in eSMC2, if the two nucleotides (DNA mutation) at one position are identical at a non-methylable site, we indicate this as 0. If the two nucleotides differ, it is indicated as 1 (i.e. a DNA mutation occurred). When assuming site-level epimutation only, three possible observations are possible at a given methylable posisiton: (1) if the two cytosines from the two chromosomes are unmethylated, it is indicated as a 2, (2) if the two cytosines are methylated, it is indicated as a 3, and (3) if at a position a cytosine is methylated and the other one unmethylated, it is indicated as a 4. Depending on the mutation, methylation and, demethylation rates, different frequencies of these states are possible in the sample of sequences, which provide information on the emission rate in the SMC method. When both site- and region-level methylation processes occur, the methylation state is conditioned by the region-level methylation state (increasing the number of possible observations to 9).

 To choose the appropriate settings for SMCm (i.e. if there are region-level epimutations), we test if the methylation state is distributed independently from one another along one genome. In the absence of a region methylation effect, the probability at each site (position) to be methylated or unmethylated should be independent of the previous position (or any other position). Conversely, if there is a region effect on epimutation, two consecutive sites along one genome would exhibit a

positive correlation in their methylated states (and across pairs of sequences). We, therefore, calculate the probability that two successive positions with an annotated methylation state would be identical under a binomial distribution of methylation along a given genome. We then compare theoretical expectations to the observed data and build the statistical test based on a binomial distribution of probabilities. If the existence of region-level epimutation is detected, the regions level methylation states are recovered through a HMM similar to *Shahryary et al., 2020*; *Denkena et al., 2021*; *Taudt et al., 2018*. Note that this HMM model does not include information from epimutation rates known from empirical studies. The complete description of the mathematical models and probabilities are in the Appendix 1 and 2.

We postulate that the epimutation rates remain unknown in most species, while the DNA mutation rate may be known (or approximated based on a closely related species). Hence, we develop an approach based on the SMC capable of leveraging information from the distribution of DNA mutations to infer the epimutation rates (similar to what is described above). Our approach first tests if epimutations violate or not the infinite site assumptions. If less than 1% of sites with their methylation state annotated are polymorphic in a sequence, we use the infinite site assumption: the site and region level epimutation rates can be recovered straightforwardly from the observed diversity ($\theta_W$, see above). Otherwise, a Baum-Welch algorithm is run to infer the most likely epimutation rates (site rate for SMP and region rates for DMRs; *van der Graaf et al., 2015*; *Vidalis et al., 2016*; *Taudt et al., 2018*).

## Calculation of the root mean square error (RMSE)

To quantify the accuracy of each demographic inference, we evaluate the root mean square error (RMSE). To do so, we choose a hundred points uniformly spread across the time window (in $\log_{10}$ scale), and compare the actual population size and the one estimated by a given method at each of these points. We thus have the following formula:

$$RMSE = \sqrt{\frac{\sum_{i=1}^{10^2} (y_i - y_i^*)^2}{10^2}}, \tag{2}$$

where $y_i$ is the true population size at the time point $i$, and $y_i^*$ is the estimated population size at the time point $i$.

## Inference of the time to the most recent common ancestor (TMRCA)

To infer the TMRCA at each position of the genome, we use an approach similar to the PSMC' described in *Schiffels and Durbin, 2014*. We first run a forward and backward algorithm on our sequence data (see appendix of *Sellinger et al., 2020*; *Terhorst et al., 2017* for computation details). From the output results, we calculate the probability to be in each hidden state at each position of the genome (note that the output product of the forward-backward algorithm is rescaled so that the sum of probability is one), which we use to compute the expected coalescent time at each position on the genome using the following formula:

$$TMRCA_i = \sum_{j=1}^{n} fo_{i,j} \times ba_{i,j} \times Tc_j, \tag{3}$$

with $i$ is the position on the genome, $j$ is the hidden state index, $n$ is the number of hidden state, $fo$ is the output from the forward algorithm, $ba$ is the output from the backward algorithm, $\sum_{j=1}^{n} fo_{i,j} \times ba_{i,j} = 1$, and $Tc$ is a vector containing all the hidden states (*i.e.* coalescent times).

## Sequence data of *Arabidopsis thaliana*

We download genome and methylome data of *A. thaliana* from the 1001 genome project *Cao et al., 2011*. We select 10 individuals from the German accessions, respectively, corresponding to the accession numbers: 9783, 9794, 9808, 9809, 9810, 9811, 9812, 9816, 9813, 9814. We only keep methylome data in CG context and in genic regions (*Vidalis et al., 2016*; *Denkena et al., 2021*). The genic regions are based on the current reference genome TAIR 10.1. The SNPs and epimutations are called according to previously published pipeline (*Taudt et al., 2018*; *Denkena et al., 2021*). As in previous studies *Sellinger et al., 2020*; *Fulgione et al., 2018*; *Durvasula et al., 2017*, we assume *A. thaliana*

data to be haploid due to high homozygosity (caused by high selfing rate). The resulting files are available on GitHub at https://github.com/TPPSellinger. To perform analysis, we chose $\mu = 6.95 \times 10^{-9}$ per generation per bp as the DNA mutation rate *Ossowski et al., 2010* and $r = 3.6 \times 10^{-8}$ as the recombination rate *Salomé et al., 2012* per generation per bp. To have the most realistic model, we assume that the methylome of *A. thaliana* undergoes both region (RMM) and site (SMM) level epimutations (*Denkena et al., 2021*). When fixed, we respectively set the site methylation and demethylation rate to $\mu_{SM} = 3.48 \times 10^{-4}$ and $\mu_{SU} = 1.47 \times 10^{-3}$ per generation per bp according to *van der Graaf et al., 2015*. We additionally set the region level methylation and demethylation rate to $\mu_{RM} = 1.6 \times 10^{-4}$ and $\mu_{RU} = 9.5 \times 10^{-4}$ per generation per bp according to *Denkena et al., 2021*. Because we do not account for the effect of variable mutation or recombination rate along the genome, we cut the five chromosomes of *A. thaliana* into eight smaller scaffolds (*Barroso and Dutheil, 2023*; *Barroso et al., 2019*). By doing this, we remove centromeric regions and limit the effect of the variation of mutation and recombination rate along the genome. The selected regions and the SNP density (from the German accessions) are represented in *Figure 6—figure supplements 2–6*.

## Acknowledgements

We thank Zhilin Zhang and Rashmi Hazarika for giving and processing the data of *Arabidopsis thaliana*. TS is supported by the Deutsche Forschungsgemeinschaft, project number 317616126 (TE809/7-1) to AT, and the Austrian Science Fund (project no. TAI 151-B) to Anja Hörger.

## Additional information

### Funding

| Funder | Grant reference number | Author |
| --- | --- | --- |
| Deutsche Forschungsgemeinschaft | 317616126 (TE809/7-1) | Aurélien Tellier |
| Austrian Science Fund | project no. TAI 151-B | Thibaut Sellinger |

The funders had no role in study design, data collection and interpretation, or the decision to submit the work for publication.

### Author contributions

Thibaut Sellinger, Conceptualization, Software, Formal analysis, Investigation, Visualization, Methodology, Writing - original draft; Frank Johannes, Conceptualization, Writing - review and editing; Aurélien Tellier, Conceptualization, Supervision, Funding acquisition, Writing - original draft, Project administration, Writing - review and editing

### Author ORCIDs

Frank Johannes ⓘ https://orcid.org/0000-0002-7962-2907
Aurélien Tellier ⓘ https://orcid.org/0000-0002-8895-0785

Reviewer #1 (Public review): https://doi.org/10.7554/eLife.89470.4.sa1
Reviewer #2 (Public review): https://doi.org/10.7554/eLife.89470.4.sa2
Author response https://doi.org/10.7554/eLife.89470.4.sa3

## Additional files

### Supplementary files

• Supplementary file 1. Supplementary Tables. (**a**) Average mean root square error (MRSE) of demographic inference in *Figure 2*, *Figure 2—figure supplement 1* and *Figure 2—figure supplement 2*. Average mean root square error (in log10) of demographic inference in *Figure 2A–D*, *Figure 2—figure supplements 1 and 2* shows the three approaches (eSMC2, SMCtheo with unknown rates, SMCtheo with known rates and MSMC2). The coefficient of variation is

indicated in parentheses (**b**) Percentage of repetitions rejecting the $H_0$ hypothesis at $P$=0.05 of binomial distribution of epimutations over 100 repetitions using two sequences of 100 Mb with recombination and mutation rate set to $1 \times 10^{-8}$ per generation per bp under a constant population size fixed to 10,000. (**c**) Average estimated rate of the site methylation and demethylation rates from simulations. True versus average estimated values of the site methylation and demethylation rates over ten repetitions. We use two sequences of 100 Mb with $r = \mu_1 = 10^{-8}$ per generation per bp under a constant population size fixed to 10,000. The coefficient of variation is indicated in brackets. (**d**) Average estimated rate of the region methylation and demethylation rates from simulations. True versus average estimated values of the region methylation and demethylation rates over ten repetitions. We use two sequences of 100 Mb with $r = \mu_1 = 10^{-8}$ per generation per bp under a constant population size fixed to 10,000. The coefficient of variation is indicated in brackets. (**e**) Average estimated rate of both site and region methylation and demethylation rates from simulations. Average estimated values of the site and region methylation and demethylation rates over ten repetitions using 2 sequences of 100 Mb with recombination and mutation rate set to $1 \times 10^{-8}$ per generation per bp under a constant population size fixed to 10,000. The coefficient of variation is indicated in brackets. (**f**) Average mean root square error of demographic inference in *Figure 5*. Average mean root square error (in log10) of demographic inference in *Figure 5* by the two approaches eSMC2, SMCm with unknown epimutations rates (A and C), and SMCm with known epimutation rates (B and D). Note the second row indicates the MRSE in recent times (younger than 400 generations ago). The coefficient of variation is indicated in parentheses (**g**) Average mean root square error of coalescent time along the genome inference. Average mean root square error of inferred coalescent time (in generation unit) along the genome over ten repetitions by the three approaches (eSMC2, SMCm with unknown epimutation rates and SMCm with known epimutation rates) under the same scenario from *Figure 5*. Inference was performed on two haploid sequences of 10 Mb with $\mu = 7 \times 10^{-9}$, $r = 3.5 \times 10^{-8}$ per generation per bp. Methylation and demethylation rates were respectively fixed to $3.5 \times 10^{-4}$ and $1.5 \times 10^{-3}$ per generation per bp. The selfing rate was fixed to 90%. The coefficient of variation is indicated in parentheses. (**h**) Average mean root square error of demographic inference in *Figure 5—figure supplements 1 and 2*. Average mean root square error (in log10) of demographic inference in *Figure 5—figure supplements 1 and 2* by the three approaches (eSMC2, SMCm with unknown epimutations rates and SMCm with known epimutation rates). Note that the second row indicates the MRSE in recent times (younger than 400 generations ago). The coefficient of variation is indicated in parentheses (**i**) Average mean root square error of demographic inference in *Figure 5—figure supplement 3*. Average mean root square error (in log10) of demographic inference in *Figure 5—figure supplement 3* by the three approaches (eSMC2, SMCm with unknown epimutations rates, SMCm with known epimutation rates). Note that the second row indicates the MRSE in recent times (younger than 400 generations ago). The coefficient of variation is indicated in parentheses (**j**) Average mean root square error of demographic inference in *Figure 5—figure supplement 4*. Average mean root square error (in log10) of demographic inference in *Figure 5—figure supplement 4* by the three approaches (eSMC2, SMCm with unknown epimutations rates, SMCm with known epimutation rates). Note that the second row indicates the MRSE in recent times (younger than 400 generations ago). The coefficient of variation is indicated in parentheses. (**k**) Average estimated rate of the site methylation and demethylation rates in *A. thaliana*. Average estimated values of the site methylation and demethylation rates by SMCm using genomes and methylomes from 10 German accessions of *A. thaliana*. We use eight scaffolds each of 10 sequences with recombination and mutation rate respectively set to $r = 3.6 \times 10^{-8}$ and $\mu_1 = 6.95 \times 10^{-9}$ per generation per bp with selfing set to 90%. The polymorphic SMPs CG sites estimations corresponds to the green line in *Figure 6*. All CG sites estimations and CG site separated by 3,000 bp corresponds to the data of the green line in *Figure 6—figure supplement 1*. (**l**) Average estimated rate of the site and region methylation and demethylation rates in *A. thaliana*. Average estimated values of the site and region methylation and demethylation rates by SMCm using genomes and methylomes from 10 German accessions of *A. thaliana*. These estimations are produced during the inference of the red line in *Figure 6—figure supplement 1*. We use eight scaffolds each of 10 sequences with recombination and mutation rate respectively set to $r = 3.6 \times 10^{-8}$ and $\mu_1 = 6.95 \times 10^{-9}$ per generation per bp with selfing set to 90%.

- MDAR checklist

## Data availability

The eSMC2 R package can be found at: https://github.com/TPPSellinger/eSMC2 (copy archived at *Sellinger, 2024a*). The input files created from *Arabidopsis thaliana* sequence data are available on GitHub at https://github.com/TPPSellinger/Arabidopsis_thaliana_methylation (copy archived at *Sellinger, 2024b*).

The following previously published dataset was used:

| Author(s) | Year | Dataset title | Dataset URL | Database and Identifier |
|---|---|---|---|---|
| Cao J, Schneeberger K, Ossowski S, Günther T, Bender S, Fitz J, Koenig D, Lanz C, Stegle O, Lippert C, Wang X, Ott F, Müller J, Borgwardt K, Schmid KJ, Weigel D, Alonso-Blanco C | 2011 | *Arabaidopsis thaliana* methylomes | https://1001genomes.org/projects/MPICao2010/index.html | Genomes, ProjectMPICao2010 |

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

## Appendix 1

### Sequentially Markovian Coalescent for several markers

Our approach is a re-implementation of the PSMC',MSMC2, eSMC and eSMC2 but accounting for different genomic markers. Hence, the Hidden Markov model is exactly the same as previously described, but with a different emission matrix. For each site, we first check what marker is present. We then set the correct substitution rate and number of hidden states. We note as *id* (identical) the event where both markers are in the same state, and *seg* if both are in different states (polymorphic). Extending the work in **Yang, 1996**, we have the following formula:

$$P(id|\gamma) = \frac{1}{nb_s} + \frac{(nb_s - 1)}{nb_s} * e^{-2\mu t_\gamma \frac{(nb_s)}{(nb_s - 1)}}$$

$$P(seg|\gamma) = \frac{(nb_s - 1)}{nb_s} - \frac{(nb_s - 1)}{nb_s} * e^{-2\mu t_\gamma \frac{(nb_s)}{(nb_s - 1)}}$$

(4)

Where $\mu$ is the substitution rate of the marker per $N$ generation, $t_\gamma$ is the average coalescent time in state $\gamma$ and $nb_s$ is the number of possible stat the marker can take.

## Appendix 2

### Sequentially Markovian Coalescent with DNA methylation (SMCm)
SMCm is similar to PSMC',MSMC2, eSMC and eSMC2 but additionally accounting for epimutations.

### Accounting only for site DNA epimutations
Here we assume epimutations occur similarly as mutations (under a finite site model). Since the model accounts for sequence and DNA methylation polymorphisms, there are at each position 5 different possible observations when comparing two sequences. The first observation is 0, corresponding to a non-methylable site where the two nucleotides are identical. 1, if the two nucleotides are different. 2 if it is a methylable site and both are unmethylated. 3, if the site is methylable and both are methylated. Finally, 4 is it a methylable site and one cytosine is methylated and the other unmethylated. Therefore, after approximating the formula assuming DNA methylation state is not affected by DNA mutations, we find:

$$P(0|\gamma) = e^{-2\mu t_\gamma}$$
$$P(1|\gamma) = 1 - e^{-2\mu t\gamma}$$
$$P(2|\gamma) = p_u \times p_{m1} \times p_{m1} + (1 - p_u) \times (1 - p_{m2}) \times (1 - p_{m2})$$
$$P(3|\gamma) = p_u \times (1 - p_{m1}) \times (1 - p_{m1}) + (1 - p_u) \times p_{m2} \times p_{m2}$$
$$P(4|\gamma) = p_u \times 2 \times p_{m1} \times (1 - p_{m1}) + (1 - p_u) \times 2 \times p_{m2} \times (1 - p_{m2})$$
$$p_u = \frac{\mu_u}{\mu_u + \mu_m}$$
$$\theta_m = (\mu_u + \mu_m) \times t_\gamma$$
$$p_{m1} = p_u + (1 - p_u) \times e^{-\theta_m}$$
$$p_{m2} = (1 - p_u) + p_u \times e^{-\theta_m}$$

(5)

Where $\mu$ is the mutation rate per nucleotide per N generation, $\mu_m$ the methylation rate per generation, $\mu_u$ the demethylation rate per generation and $t_\gamma$ the average coalescent time in state γ. Additionally, we define $p_u$ as the probability to be unmethylated at equilibrium, as well as $p_{m1}$ and $p_{m2}$ the respective probability to stay unmethylated or methylated after a time $t_\gamma$.

### Accounting only for region epimutations
Here, we assume region epimutation occurs similarly to mutations (under a finite site model). However, unlike previously, the epimutations affect multiple sites; hence, only the first position of a methylated region is considered, and the following positions will be considered as missing data (because it is one block of information). Therefore, there are at each position 6 different possible observations when comparing two sequences. The first observation is 0, corresponding to a non-methylable site where the two nucleotides are identical. 1, if the two nucleotides are different. 2 if it is a region with methylation state annotated and both regions are unmethylated. 3, if it is a region with methylation state annotated and both regions are methylated. 4 if it is a region with methylation state annotated and both regions are in different methylation states. 5 is missing data. Therefore, after approximating the formula, assuming the methylation state is not affected by mutations, we have the following formula:

$$P(0|\gamma) = e^{-2\mu t_\gamma}$$

$$P(1|\gamma) = 1 - e^{-2\mu t_\gamma}$$

$$P(2|\gamma) = p_u \times p_{m1} \times p_{m1} + (1 - p_u) \times (1 - p_{m2}) \times (1 - p_{m2})$$

$$P(3|\gamma) = p_u \times (1 - p_{m1}) \times (1 - p_{m1}) + (1 - p_u) \times p_{m2} \times p_{m2}$$

$$P(4|\gamma) = p_u \times 2 \times p_{m1} \times (1 - p_{m1}) + (1 - p_u) \times 2 \times p_{m2} \times (1 - p_{m2})$$

$$P(5|\gamma) = 1 \qquad\qquad (6)$$

$$p_u = \frac{\mu_u}{\mu_u + \mu_m}$$

$$\theta_m = (\mu_u + \mu_m) \times t_\gamma$$

$$p_{m1} = p_u + (1 - p_u) \times e^{-\theta_m}$$

$$p_{m2} = (1 - p_u) + p_u \times e^{-\theta_m}$$

Where $\mu$ is the mutation rate per nucleotide per N generation, $\mu_m$ the region methylation rate per generation, $\mu_u$ the region demethylation rate per generation and $t_\gamma$ the average coalescent time in state $\gamma$. Additionally, we define $p_u$ as the probability for the region to be unmethylated at equilibrium, as well as $p_{m1}$ and $p_{m2}$ the respective probability for the region to stay unmethylated or methylated after a time $t_\gamma$. To recover the region epimutations, we use a hidden Markov model (HMM). The HMM takes as input genome and methylome data described above and compares two sequences to recover epiregion. The hidden Markov model has 9 hidden states: 1 (regions with no methylation information), 2 (no methylation information in individual 1& mainly methylated region in individual 2), 3 (no methylation information in individual 1& mainly unmethylated region in individual 2), 4 (no methylation information in individual 2& mainly methylated region in individual 1), 5 (mainly methylated region in individual 1& mainly methylated region in individual 2), 6 (mainly methylated region in individual 1& mainly unmethylated region in individual 2), 7 (mainly unmethylated region in individual 1& no methylation information in individual 2), 8 (mainly unmethylated region in individual 1& mainly methylated region in individual 2), 9 (mainly unmethylated region in individual 1& mainly unmethylated region in individual 2). Additional parameters are necessary to define the transition rate. The user needs to define the minimum number of annotated methylable sites to form a region (by default 4) and minimum size of a region in bp (100 by default). Our approach then defines the transition rate as the transition rate maximizing the number of regions respecting the defined criteria. The emission matrix is defined by the user, but by default, the probability of observing methylated and unmethylated in a methylated region is set to 0.8 and 0.2 respectively. In an unmethylated region, the probabilities are respectively set to 0.2 and 0.8. We use those values throughout the study.

## Accounting for site and region epimutations

Here, we assume site and region epimutation occur similarly to mutations (under a finite site model). Like previously, the epimutations affect multiple sites; hence, only the first position of a methylated region is considered, and the following positions will be considered as missing data (because it is one block of information). However, the observation will depend on the region. Therefore, there are at each position 10 different possible observations when comparing two sequences. The first observation is 0, corresponding to a non-methylable site where the two nucleotides are identical. 1, if the two nucleotides are different. 2 if it is a region with methylation state annotated and both regions are unmethylated, and all sites are unmethylated. 3 if it is a region with methylation state annotated, both regions are unmethylated, and all sites are methylated. 4 if it is a region with methylation state annotated, and both regions are unmethylated, and at least one site is segregating. 5 if it is a region with methylation state annotated and both regions are methylated, and all sites are unmethylated. 6 if it is a region with a methylation state annotated, both regions are methylated, and all sites are methylated. 7 if it is a region with methylation state annotated, both regions are methylated, and at least one site is segregating. 8 if it is a region with a methylation state annotated and both regions are in different methylation states. 9 is missing data. Therefore, we have the following formula:

$$P(0|\gamma) = e^{-2\mu t\gamma}$$

$$P(1|\gamma) = 1 - e^{-2\mu t\gamma}$$

$$P(2|\gamma) = (P_O(P_{OO}P_{OO})) + (P_P(P_{PO}P_{PO})) + (P_M(P_{MO}P_{MO})) + (P_U(P_{UO}P_{UO}))$$

$$P(3|\gamma) = (P_O(P_{OP}P_{OP})) + (P_P(P_{PP}P_{PP})) + (P_M(P_{MP}P_{MP})) + (P_U(P_{UP}P_{UP}))$$

$$P(4|\gamma) = (P_O(2P_{OP}P_{OO})) + (P_P(2P_{PP}P_{PO})) + (P_M(2P_{MP}P_{MO})) + (P_U(2P_{UP}P_{UO}))$$

$$P(5|\gamma) = (P_O(P_{OU}P_{OU})) + (P_P(P_{PU}P_{PU})) + (P_M(P_{MU}P_{MU})) + (P_U(P_{UU}P_{UU}))$$ \quad (7)

$$P(6|\gamma) = (P_O(P_{OM}P_{OM})) + (P_P(P_{PM}P_{PM})) + (P_M(P_{MM}P_{MM})) + (P_U(P_{UM}P_{UM}))$$

$$P(7|\gamma) = (P_O(2P_{OU}P_{OM})) + (P_P(2P_{PU}P_{PM})) + (P_M(2P_{MU}P_{MM})) + (P_U(2P_{UU}P_{UM}))$$

$$P(8|\gamma) = (p_u 2p_{m1}(1 - p_{m1})) + (1 - p_u)2p_{m2}(1 - p_{m2})$$

$$P(9|\gamma) = 1$$

Where we have:

$$p_u = \frac{\mu_u}{\mu_u + \mu_m}; p_{reg\_d} = \frac{\mu_{reg\_d}}{\mu_{reg\_d} + \mu_{reg\_m}}$$

$$\theta_m = (\mu_u + \mu_m)t_\gamma; \theta_{reg\_m} = (\mu_{reg\_d} + \mu_{reg\_m})t_\gamma$$

$$p_{m1} = (p_u + ((1 - p_u)e^{(-\theta_m)})); p_{m2} = ((1 - p_u) + (p_u e^{(-\theta_m)}))$$

$$p_{reg\_m1} = (p_{reg\_d} + ((1 - p_{reg\_d})e^{(-\theta_{reg\_m})})); p_{reg\_m2} = ((1 - p_{reg\_d}) + (p_{reg\_d}e^{(-\theta_{reg\_m})}))$$

$$t_m = \frac{1}{(\mu_{reg\_d} + \mu_{reg\_m})}$$

$$p_{eq1} = (p + ((1 - p)e^{(-(\mu_u+\mu_m)*(t_m))})); p_{eq2} = ((1 - p) + (pe^{(-(\mu_u+\mu_m)*(t_m))}))$$

$$P_{no\_reg\_event} = (exp(-(\mu_{reg\_d} + \mu_{reg\_m}) * t_\gamma))$$

$$p_{m1\_small} = (p_u + ((1 - p_u)e^{(-(\mu_u+\mu_m)*((\frac{1}{(\mu_{reg\_d} + \mu_{reg\_m})}) - (\frac{(t_\gamma e^{(-(\mu_{reg\_d}+\mu_{reg\_m})t_\gamma)})}{(1 - exp(-(\mu_{reg\_d} + \mu_{reg\_m})t_\gamma))})))}))$$

$$p_{m2\_small} = ((1 - p_u) + (p_u e^{(-(\mu_u+\mu_m)*((\frac{1}{(\mu_{reg\_d} + \mu_{reg\_m})}) - (\frac{(t_\gamma e^{(-(\mu_{reg\_d}+\mu_{reg\_m})t_\gamma)})}{(1 - exp(-(\mu_{reg\_d} + \mu_{reg\_m})t_\gamma))})))}))$$

$$P_O = (p_{reg\_d})(1 - p_{eq1})$$

$$P_P = (p_{reg\_d})(p_{eq1})$$

$$P_U = (1 - p_{reg\_d})(1 - p_{eq2})$$

$$P_M = (1 - p_{reg\_d})(p_{eq2})$$

$$P_{OO} = p_{reg\_m1}(((1 - P_{no\_reg\_event})(1 - p_{m1\_small})) + (P_{no\_reg\_event}(p_{m2})))$$

$$P_{OP} = p_{reg\_m1}(((1 - P_{no\_reg\_event})(p_{M1_small})) + (P_{no\_reg\_event}(1 - p_{m2})))$$

$$P_{OM} = (1 - p_{reg\_m1})p_{m2\_small}$$

$$P_{OU} = (1 - p_{reg\_m1})(1 - p_{m2\_small})$$

$$P_{PO} = p_{reg\_m1}(((1 - P_{no\_reg\_event})(1 - p_{m1\_small})) + (P_{no\_reg\_event}(1 - p_{m1})))$$

$$P_{PP} = p_{reg\_m1}(((1 - P_{no\_reg\_event})(p_{m1\_small})) + (P_{no\_reg\_event}(p_{m1})))$$

$$P_{PM} = (1 - p_{reg\_m1})p_{m2\_small}$$

$$P_{PU} = (1 - p_{reg\_m1})(1 - p_{m2\_small})$$

$$P_{MO} = (1 - p_{reg\_m2})(1 - p_{m1\_small})$$

$$P_{MP} = (1 - p_{reg\_m2})(p_{m1\_small})$$

$$P_{MM} = p_{reg\_m2}(((1 - P_{no\_reg\_event})(p_{m2\_small})) + (P_{no\_reg\_event}(p_{m2})))$$

$$P_{MU} = p_{reg\_m2}(((1 - P_{no\_reg\_event})(1 - p_{m2\_small})) + (P_{no\_reg\_event}(1 - p_{m2})))$$

$$P_{UO} = (1 - p_{reg\_m2})(1 - p_{m1\_small})$$

$$P_{UP} = (1 - p_{reg\_m2})(p_{m1\_small})$$

$$P_{UM} = p_{reg\_m2}(((1 - P_{no\_reg\_event})(p_{m2\_small})) + (P_{no\_reg\_event}(1 - p_{m1})))$$

$$P_{UU} = p_{reg\_m2}(((1 - P_{no\_reg\_event})(1 - p_{m2\_small})) + (P_{no\_reg\_event}(p_{m1})))$$

(8)

Where $\mu$ is the mutation rate per nucleotide per N generation, $\mu_m$ the site methylation rate per generation, $\mu_u$ the site demethylation rate, $\mu_{reg\_m}$ the region methylation rate per generation, $\mu_{reg\_d}$ the region demethylation rate per generation and $t_\gamma$ the average coalescent time in state $\gamma$. Please refer to the R function build_emi_m from the Package eSMC2 for a clearer description of the probabilities. Note that in the text, these probabilities are written as $\mu_1$ for mutation rate per nucleotide, $\mu_{SM}$ the site methylation rate, $\mu_{SU}$ the site demethylation rate, $\mu_{RM}$ the region methylation rate, and $\mu_{RU}$ the region demethylation rate.

