## [Editor Report · eLife assessment]

This **important** study extends existing sequentially Markovian coalescent approaches to include the combined use of SNPs and hypervariable loci such as epimutations. This is an intriguing addition to infer population size history in the recent past, and the authors provide **solid** validation of their methods via simulation and analysis of empirical data in *Arabidopsis thaliana*. Given the increasing availability of such data, this work is a timely contribution and represents a foundation for further developments to explore when and where these methods will be best used.

---

## [Referee Report · Reviewer #1 (Public review)]

The authors developed an extension to the pairwise sequentially Markov coalescent model that allows to simultaneously analyze multiple types of polymorphism data. In this paper, they focus on SNPs and DNA methylation data. Since methylation markers mutate at a much faster rate than SNPs, this potentially gives the method better power to infer size history in the recent past. Additionally, they explored a model where there are both local and regional epimutational processes.

Integrating additional types of heritable markers into SMC is a nice idea which I like in principle. However, a major caveat to this approach seems to be a strong dependence on knowing the epimutation rate. In Fig. 6 it is seen that, when the epimutation rate is known, inferences do indeed look better; but this is not necessarily true when the rate is not known. (See also major comment #1 below about the interpretation of these plots.) A roughly similar pattern emerges in Supp. Figs. 4-7; in general, results when the rates have to be estimated don't seem that much better than when focusing on SNPs alone. This carries over to the real data analysis too: the interpretation in Fig. 7 appears to hinge on whether the rates are known or estimated, and the estimated rates differ by a large amount from earlier published ones.

Overall, this is an interesting research direction, and I think the method may hold more promise as we get more and better epigenetic data, and in particular better knowledge of the epigenetic mutational process.

---

## [Referee Report · Reviewer #2 (Public review)]

A limitation in using SNPs to understand recent histories of genomes is their low mutation frequency. Tellier et al. explore the possibility of adding hypermutable markers to SNP based methods for better resolution over short time frames. In particular, they hypothesize that epimutations (CG methylation and demethylation) could provide a useful marker for this purpose. Individual CGs in Arabidopsis tends to be either close to 100% methylated or close to 0%, and are inherited stably enough across generations that they can be treated as genetic markers. Small regions containing multiple CGs can also be treated as genetic markers based on their cumulative methylation level. In this manuscript, Tellier et al develop computational methods to use CG methylation as a hypermutable genetic marker and test them on theoretical and real data sets. They do this both for individual CGs and small regions. My review is limited to the simple question of whether using CG methylation for this purpose makes sense at a conceptual level, not at the level of evaluating specific details of the methods. I have a small concern in that it is not clear that CG methylation measurements are nearly as binary in other plants and other eukaryotes as they are in Arabidopsis. However, I see no reason why the concept of this work is not conceptually sound. Especially in the future as new sequencing technologies provide both base calling and methylating calling capabilities, using CG methylation in addition to SNPs could become a useful and feasible tool for population genetics in situations where SNPs are insufficient.

---

## [Author Response]

The following is the authors’ response to the previous reviews.

**Public Reviews:**

**Reviewer #1 (Public Review):**
The authors developed an extension to the pairwise sequentially Markov coalescent model that allows to simultaneously analyze multiple types of polymorphism data. In this paper, they focus on SNPs and DNA methylation data. Since methylation markers mutate at a much faster rate than SNPs, this potentially gives the method better power to infer size history in the recent past. Additionally, they explored a model where there are both local and regional epimutational processes. Integrating additional types of heritable markers into SMC is a nice idea which I like in principle. However, a major caveat to this approach seems to be a strong dependence on knowing the epimutation rate. In Fig. 6 it is seen that, when the epimutation rate is known, inferences do indeed look better; but this is not necessarily true when the rate is not known. (See also major comment #1 below about the interpretation of these plots.) A roughly similar pattern emerges in Supp. Figs. 4-7; in general, results when the rates have to be estimated don't seem that much better than when focusing on SNPs alone. This carries over to the real data analysis too: the interpretation in Fig. 7 appears to hinge on whether the rates are known or estimated, and the estimated rates differ by a large amount from earlier published ones.Overall, this is an interesting research direction, and I think the method may hold more promise as we get more and better epigenetic data, and in particular better knowledge of the epigenetic mutational process. At the same time, I would be careful about placing too much emphasis on new findings that emerge solely by switching to SNP+SMP analysis.Major comments:- For all of the simulated demographic inference results, only plots are presented. This allows for qualitative but not quantitative comparisons to be made across different methods. It is not easy to tell which result is actually better. For example, in Supp. Fig. 5, eSMC2 seems slightly better in the ancient past, and times the trough more effectively, while SMCm seems a bit better in the very recent past. For a more rigorous approach, it would be useful to have accompanying tables that measure e.g. mean-squared error (along with confidence intervals) for each of the different scenarios, similar to what is already done in Tables 1 and 2 for estimating $r$.

We believe this comment was addressed in the previous revision (Sup Table 6-10) by adding Root Mean Square Errors for the demographic estimates (and RMSE for recent versus past portions of the demography).

- 434: The discussion downplays the really odd result that inputting the true value of the mutation rate, in some cases, produces much worse estimates than when they are learned from data (SFig. 6)! I can't think of any reason why this should happen other than some sort of mathematical error or software bug. I strongly encourage the authors to pin down the cause of this puzzling behaviour. (Comment addressed in revision. Still, I find the explanation added at 449ff to be somewhat puzzling -- shouldn't the results of the regional HMM scan only improve if the true mutation rate is given?)

We do understand that our results and explanation can appear counter-intuitive. As acknowledged by the reviewer, in the previous round of revision we have at length clarified this puzzling behaviour by the discrepancy in assessing methylation regions using the HMM method which then differs from the HMM for the SMC inference. We are happy to clarify further in response to the new question of reviewer 1:

If the Reviewer #1 means the SNP mutations (e.g. A → T), knowing the true mutation rate does not help the HMM to recover the region level methylation status.

If the Reviewer #1 means the epimutations (whether it is the region, site or both), knowing the true epimutations rates could theoretically help the HMM to recover the region level methylation status. However, at present, our method does not leverage information from epimutation rates to infer the region level methylation status. As inferring the epimutations rates is one of the goals of this study in the SMC inference, and that region level methylation status is required to infer those rates, we suspect that using epimutations rates to infer the region level methylation status could be statistically inappropriate (generating some kind of circular estimations). Instead, our HMM uses only the proportion of methylated and unmethylated sites (estimated from the genome) to determine whether or not a region status is most-likely to be methylated or unmethylated. We now explicit this fact in the HMM for methylation region in the method section.

We acknowledge that our HMM to infer region level methylation status could be improved, but this would be a complete project and study on its own (due to the underlying complexity of the finite site and the lack of a consensus model for epimutations at evolutionary time scale). We believe our HMM to have been the best compromise with what was known from methylation and our goals when the study was conducted, and future work is definitely worth conducting on the estimation of the methylation regions.

- As noted at 580, all of the added power from integrating SMPs/DMRs should come from improved estimation of recent TMRCAs. So, another way to study how much improvement there is would be to look at the true vs. estimated/posterior TMRCAs. Although I agree that demographic inference is ultimately the most relevant task, comparing TMRCA inference would eliminate other sources of differences between the methods (different optimization schemes, algorithmic/numerical quirks, and so forth). This could be a useful addition, and may also give you more insight into why the augmented SMC methods do worse in some cases. (Comment addressed in revision via Supp. Table 7.).- A general remark on the derivations in Section 2 of the supplement: I checked these formulas as best I could. But a cleaner, less tedious way of calculating these probabilities would be to express the mutation processes as continuous time Markov chains. Then all that is needed is to specify the rate matrices; computing the emission probabilities needed for the SMC methods reduces to manipulating the results of some matrix exponentials. In fact, because the processes are noninteracting, the rate matrix decomposes into a Kronecker sum of the individual rate matrices for each process, which is very easy to code up. And this structure can be exploited when computing the matrix exponential, if speed is an issue.

We believe this comment was acknowledged in the previous revision (line 649), and we thank the reviewer for this interesting insight.

- Most (all?) of the SNP-only SMC methods allow for binning together consecutive observations to cut down on computation time. I did not see binning mentioned anywhere, did you consider it? If the method really processes every site, how long does it take to run?

We believe this comment was addressed in the previous revision and was added to the manuscript in the methods Section (subsection : SMC optimization function).

- 486: The assumed site and region (de)methylation rates listed here are several OOM different from what your method estimated (Supp. Tables 5-6). Yet, on simulated data your method is usually correct to within an order of magnitude (Supp. Table 4). How are we to interpret this much larger difference between the published estimates and yours? If the published estimates are not reliable, doesn't that call into question your interpretation of the blue line in Fig. 7 at 533? (Comment addressed in revision.)
**Reviewer #2 (Public Review):**
A limitation in using SNPs to understand recent histories of genomes is their low mutation frequency. Tellier et al. explore the possibility of adding hypermutable markers to SNP based methods for better resolution over short time frames. In particular, they hypothesize that epimutations (CG methylation and demethylation) could provide a useful marker for this purpose. Individual CGs in Arabidopsis tends to be either close to 100% methylated or close to 0%, and are inherited stably enough across generations that they can be treated as genetic markers. Small regions containing multiple CGs can also be treated as genetic markers based on their cumulative methylation level. In this manuscript, Tellier et al develop computational methods to use CG methylation as a hypermutable genetic marker and test them on theoretical and real data sets. They do this both for individual CGs and small regions. My review is limited to the simple question of whether using CG methylation for this purpose makes sense at a conceptual level, not at the level of evaluating specific details of the methods. I have a small concern in that it is not clear that CG methylation measurements are nearly as binary in other plants and other eukaryotes as they are in Arabidopsis. However, I see no reason why the concept of this work is not conceptually sound. Especially in the future as new sequencing technologies provide both base calling and methylating calling capabilities, using CG methylation in addition to SNPs could become a useful and feasible tool for population genetics in situations where SNPs are insufficient.

We thank again the reviewer #2 for his positive comments.

**Reviewer #3 (Public Review):**
I very much like this approach and the idea of incorporating hypervariable markers. The method is intriguing, and the ability to e.g. estimate recombination rates, the size of DMRs, etc. is a really nice plus. I am not able to comment on the details of the statistical inference, but from what I can evaluate it seems reasonable and in principle the inclusion of highly mutable sties is a nice advance. This is an exciting new avenue for thinking about inference from genomic data. I remain a bit concerned about how well this will work in systems where much less is understood about methylation,The authors include some good caveats about applying this approach to other systems, but I think it would be helpful to empiricists outside of thaliana or perhaps mammalian systems to be given some indication of what to watch out for. In maize, for example, there is a nonbimodal distribution of CG methlyation (35% of sites are greater than 10% and less than 90%) but this may well be due to mapping issues. The authors solve many of the issues I had concerns with by using gene body methylation, but this is only briefly mentioned on line 659. I'm assuming the authors' hope is that this method will be widely used, and I think it worth providing some guidance to workers who might do so but who are not as familiar with these kind of data.

We thank the reviewer #3 for his positive comments. And we agree with Reviewer #3 concerning the application to data and that our approach needs to be carefully thought before applied. Our results clearly show that methylation processes are not well enough understood to apply our approach as we initially (maybe naively) designed it. Further investigations need to be conducted and appropriate theoretical models need to be developed before reliable results can be obtained. And we hope that our discussion points this out. However, our approach, the theoretical models and the additional tools contained in this study can be used to help researchers in their investigations to whether or not use different genomic markers to build a common (potentially more reliable) ancestral history. We enhanced the discussion in this second revision by clarifying also the use of the methylation from genic regions to avoid confusion (lines 700-731).

**Recommendations for the authors:**

**Reviewer #1 (Recommendations For The Authors):**
In added Supp. Table 7, I don't think these are in log10 units as stated in the caption.

Well Spotted! Indeed, the RMSE is not in log10 scale, we corrected the caption. We also added that the TMRCA used for MRSE calculations is in generations units to avoid potential confusion.

**Reviewer #3 (Recommendations for The Authors):**
I very much appreciate the authors' attention to previous questions. I would ask that a bit more is spent in the discussion on concerns/approaches empiricists should keep in mind -- I am wary of this being uncritically applied to data from non-model species. It was not clear to me, for example (only mentioned on line 659 in the discussion) that the thaliana data is only using gene-body methylation. This poses potential issues with background selection that the authors acknowledge appropriately, but also assuages many of my concerns about using genome-wide data. I think text with recommendations for data/filtering/etc or at least cautions of assumptions empiricists should be aware of would help.

We apologize for the confusion at line 659. As written in the other section of the manuscript we meant CG sites in genic regions (and not only gene body methylated regions).

Due to the manuscript’s structure, the data from *Arabidopsis thaliana* is only described at the very end of the manuscript (line 900+). However, a brief description could also be found line 291-296. We however added a sentence in the introduction (line 128) for clarity.

We however agree with the comment made by reviewer #3 concerning the application to data. We pointed in the discussion the risk of applying our approach on ill-understood (or illprepared) data and stressed the current need of studies on the epimutations processes at evolutionary time scale (i.e. at Ne time scale) (line 700-703).